# HyAR: Addressing Discrete-Continuous Action Reinforcement Learning via Hybrid Action Representation

**Boyan Li**[1*], **Hongyao Tang**[1*], **Yan Zheng**[1†], **Jianye Hao**[1†], **Pengyi Li**[1], **Zhen Wang**[2†],
**Zhaopeng Meng**[1], **Li Wang**[1]
[1]College of Intelligence and Computing, Tianjin University,
[2]School of Artificial Intelligence, Optics and Electronics (iOPEN) and School of
Cybersecurity, Northwestern Polytechnical University

## Abstract

Discrete-continuous hybrid action space is a natural setting in many practical problems, such as robot control and game AI. However, most previous Reinforcement Learning (RL) works only demonstrate the success in controlling with either discrete or continuous action space, while seldom take into account the hybrid action space. One naive way to address hybrid action RL is to convert the hybrid action space into a unified homogeneous action space by discretization or continualization, so that conventional RL algorithms can be applied. However, this ignores the underlying structure of hybrid action space and also induces the scalability issue and additional approximation difficulties, thus leading to degenerated results. In this paper, we propose **Hy**brid **A**ction **R**epresentation (**HyAR**) to learn a compact and decodable latent representation space for the original hybrid action space. HyAR constructs the latent space and embeds the dependence between discrete action and continuous parameter via an embedding table and conditional Variational Auto-Encoder (VAE). To further improve the effectiveness, the action representation is trained to be semantically smooth through unsupervised environmental dynamics prediction. Finally, the agent then learns its policy with conventional DRL algorithms in the learned representation space and interacts with the environment by decoding the hybrid action embeddings to the original action space. We evaluate HyAR in a variety of environments with discrete-continuous action space. The results demonstrate the superiority of HyAR when compared with previous baselines, especially for high-dimensional action spaces.

## 1 Introduction

Deep Reinforcement learning (DRL) has recently shown a great success in a variety of decision-making problems that involve controls with either discrete actions, such as Go (Silver et al., 2016) and Atari (Mnih et al., 2015), or continuous actions, such as robot control (Schulman et al., 2015; Lillicrap et al., 2015). However, in contrast to these two kinds of homogeneous action space, many real-world scenarios requires more complex controls with discrete-continuous hybrid action space, e.g., Robot Soccer (Masson et al., 2016) and Games (Xiong et al., 2018). For example, in robot soccer, the agent not only needs to choose whether to shoot or pass the ball (i.e., discrete actions) but also the associated angle and force (i.e., continuous parameters). Such a hybrid action is also called parameterized action in some previous works (Hausknecht & Stone, 2016; Massaroli et al., 2020). Unfortunately, most conventional RL algorithms cannot deal with such a heterogeneous action space directly, thus preventing the application of RL in these kinds of practical problems.

To deal with hybrid action space, the most straightforward approach is to convert the heterogeneous space into a homogeneous one through discretization or continualization. However, it is apparent

---

*Equal contribution.
†Corresponding authors: Yan Zheng (yanzheng@tju.edu.cn) , Jianye Hao (jianye.hao@tju.edu.cn) and Zhen Wang(w-zhen@nwpu.edu.cn).

that discretizing continuous parameter space suffers from the scalability issue due to the exponentially exploring number of discretized actions; while casting all discrete actions into a continuous dimension produces a piecewise-function action subspace, resulting in additional difficulties in approximation and generalization. To overcome these problems, a few recent works propose specific policy structures to learn DRL policies over the original hybrid action space directly. Parameterized Action DDPG (PADDPG) (Hausknecht & Stone, 2016) makes use of a DDPG (Lillicrap et al., 2015) structure where the actor is modified to output a unified continuous vector as the concatenation of values for all discrete actions and all corresponding continuous parameters. By contrast, Hybrid PPO (HPPO) (Fan et al., 2019) uses multiple policy heads consisting of one for discrete actions and the others for corresponding continuous parameter of each discrete action separately. A very similar idea is also adopted in (Peng & Tsuruoka, 2019). These methods are convenient to implement and are demonstrated to effective in simple environments with low-dimensional hybrid action space. However, PADDPG and HPPO neglect the dependence between discrete and continuous components of hybrid actions, thus can be problematic since the dependence is vital to identifying the optimal hybrid actions in general. Besides, the modeling of all continuous parameter dimensions all the time introduces redundancy in computation and policy learning, and may also have the scalability issue when the hybrid action space becomes high-dimensional.

To model the dependence, Parameterized DQN (PDQN) (Xiong et al., 2018) proposes a hybrid structure of DQN (Mnih et al., 2015) and DDPG. The discrete policy is represented by a DQN which additionally takes as input all the continuous parameters output by the DDPG actor; while the DQN also serves as the critic of DDPG. Some variants of such a hybrid structure are later proposed in (Delalleau et al., 2019; Ma et al., 2021; Bester et al.,

| Algorithm | Scalability | Stationarity | Dependence | Latent |
|-----------|:-----------:|:------------:|:----------:|:------:|
| PADDPG | ✗ | ✓ | ✗ | ✗ |
| HPPO | ✗ | ✓ | ✗ | ✗ |
| PDQN | ✗ | ✓ | ✓ | ✗ |
| HHQN | ✓ | ✗ | ✓ | ✗ |
| HyAR (Ours) | ✓ | ✓ | ✓ | ✓ |

Table 1: A comparison on algorithmic properties of existing methods for discrete-continous hybrid action RL.

2019). Due to the DDPG actor's modeling of all parameters, PDQN also have the redundancy and potential scalablity issue. In an upside-down way, Hierarchical Hybrid Q-Network (HHQN) (Fu et al., 2019) models the dependent hybrid-action policy with a two-level hierarchical structure. The high level is for the discrete policy and the selected discrete action serves as the condition (in analogy to subgoal) which the low-level continuous policy conditions on. This can be viewed as a special two-agent cooperative game where the high level and low level learn to coordinate at the optimal hybrid actions. Although the hierarchical structure (Wei et al., 2018) seems to be natural, it suffers from the high-level non-stationarity caused by off-policy learning dynamics (Wang et al., 2020), i.e., a discrete action can no longer induce the same transition in historical experiences due to the change of the low-level policy. In contrast, PADDPG, HPPO and PDQN are stationary in this sense since they all learn an overall value function and policy thanks to their special structures, which is analogous to learning a joint policy in the two-agent game. All the above works focus on policy learning over original hybrid action space. As summarized in Table 1, none of them is able to offer three desired properties, i.e., scalability, stationarity and action dependence, at the same time.

In this paper, we propose a novel framework for hybrid action RL, called **Hy**brid **A**ction **R**epresentation (**HyAR**), to achieve all three properties in Table 1. A conceptual overview of HyAR is shown in Fig. 1. The main idea is to construct a unified and decodable representation space for original discrete-continuous hybrid actions, among which the agent learns a latent policy. Then, the selected latent action is decoded back to the original hybrid action space so as to interact with the environment. HyAR is inspired by recent advances in Representation Learning in DRL. Action representation learning has shown the potentials in boosting learning performance (Whitney et al., 2020), reducing large discrete action space (Chandak et al., 2019), improving generalization in offline RL (Zhou et al., 2020) and so on. Different from these works, to the best knowledge, we are the

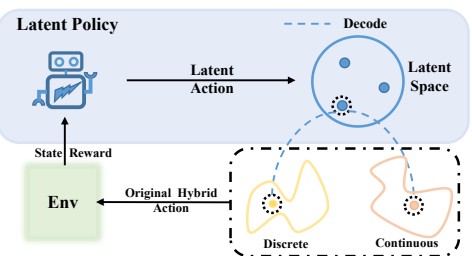

Figure 1: An overview of HyAR. Agent learns a latent policy in the latent representation space of discrete-continuous actions. The selected latent action is then decoded into the original space so as to interact with the environment.

first to propose representation learning for discrete-continuous hybrid actions, which consist of heterogeneous and dependent action components.

In HyAR, we maintain a continuous vector for each discrete action in a learnable embedding table; then a conditional Variational Auto-encoder (VAE) (Kingma & Welling, 2014) that conditions on the state and the embedding of discrete action is used to construct the latent representation space for the associated continuous parameters. Different from HHQN, the conditional VAE models and embeds the dependence in an implicit fashion. The learned representation space are compact and thus scalable, while also provides convenient decoding by nearest-neighbor lookup of the embedding table and the VAE decoder. Moreover, we utilize the unsupervised environmental dynamics to learn dynamics predictive hybrid action representation. Such a representation space can be semantically smooth, i.e., hybrid action representations that are close in the space have similar influence on environmental dynamics, thus benefits hybrid action RL in the representation space. With the constructed action representation space, we use TD3 algorithm (Fujimoto et al., 2018) for the latent policy learning. To ensure the effectiveness, we further propose two mechanisms: *latent space constraint* and *representation shift correction* to deal with unreliable latent representations and outdated off-policy action representation experiences respectively. In our experiments, we evaluate HyAR in a few representative environments with hybrid action space, as well as several new and more challenging benchmarks.

Our main contributions are summarized below:

- We propose a novel and generic framework for discrete-continuous hybrid action RL by leveraging representation learning of hybrid action space for the first time.
- We propose an unsupervised method of learning a compact and decodable representation space for discrete-continuous hybrid actions, along with two mechanisms to improve the effectiveness of latent policy learning.
- Our algorithm consistently outperforms prior algorithms in representative hybrid-action benchmarks, especially demonstrating significant superiority when the hybrid action space becomes larger. For reproducibility, codes are provided in the supplementary material.

## 2 PRELIMINARIES

**Markov Decision Process** Consider a standard Markov Decision Process (MDP) $\langle \mathcal{S}, \mathcal{A}, \mathcal{P}, \mathcal{R}, \gamma, T \rangle$, defined with a state set $\mathcal{S}$, an action set $\mathcal{A}$, transition function $\mathcal{P} : \mathcal{S} \times \mathcal{A} \times \mathcal{S} \to \mathbb{R}$, reward function $\mathcal{R} : \mathcal{S} \times \mathcal{A} \to \mathbb{R}$, discounted factor $\gamma \in [0, 1)$ and horizon $T$. The agent interacts with the MDP by performing its policy $\pi : \mathcal{S} \to \mathcal{A}$. The objective of an RL agent is to optimize its policy to maximize the expected discounted cumulative reward $J(\pi) = \mathbb{E}_\pi[\sum_{t=0}^{T} \gamma^t r_t]$, where $s_0 \sim \rho_0(s_0)$ the initial state distribution, $a_t \sim \pi(s_t)$, $s_{t+1} \sim \mathcal{P}(s_{t+1} \mid s_t, a_t)$ and $r_t = \mathcal{R}(s_t, a_t)$. The state-action value function $Q^\pi$ is defined as $Q^\pi(s, a) = \mathbb{E}_\pi\left[\sum_{t=0}^{T} \gamma^t r_t \mid s_0 = s, a_0 = a\right]$.

**Parameterized Action MDP** In this paper, we focus on a Parameterized Action Markov Decision Process (PAMDP) $\langle \mathcal{S}, \mathcal{H}, \mathcal{P}, \mathcal{R}, \gamma, T \rangle$ (Masson et al., 2016). PAMDP is an extension of stardard MDP with a discrete-continuous hybrid action space $\mathcal{H}$ defined as:

$$\mathcal{H} = \{(k, x_k) \mid x_k \in \mathcal{X}_k \text{ for all } k \in \mathcal{K}\}, \tag{1}$$

where $\mathcal{K} = \{1, \cdots, K\}$ is the discrete action set, $\mathcal{X}_k$ is the corresponding continuous parameter set for each $k \in \mathcal{K}$. We call any pair of $k, x_k$ as a hybrid action and also call $\mathcal{H}$ as hybrid action space for short in this paper. In turn, we have state transition function $\mathcal{P} : \mathcal{S} \times \mathcal{H} \times \mathcal{S} \to \mathbb{R}$, reward function $\mathcal{R} : \mathcal{S} \times \mathcal{H} \to \mathbb{R}$, agent's policy $\pi : \mathcal{S} \to \mathcal{H}$ and hybrid-action value function $Q^\pi(s, k, x_k)$.

Conventional RL algorithms are not compatible with hybrid action space $\mathcal{H}$. Typical policy representations such as Multinomial distribution or Gaussian distribution can not model the heterogeneous components among the hybrid action. Implicit policies derived by action value functions, often adopted in value-based algorithms, also fail due to intractable maximization over infinite hybrid actions. In addition, there exists the dependence between discrete actions and continuous parameters, as a discrete action $k$ determines the valid parameter space $\mathcal{X}_k$ associated with it. In other words, the same parameter paired with different discrete actions can be significantly different in semantics. This indicate that in principle an optimal hybrid-action policy can not determine the continuous parameters beforehand the discrete action is selected.

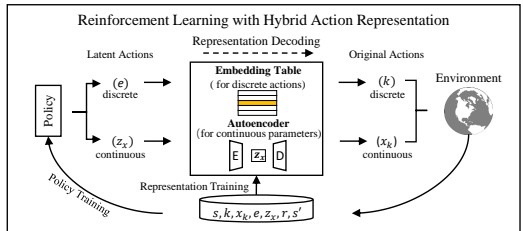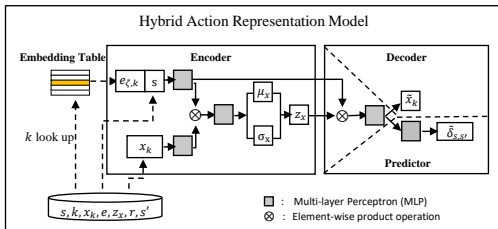

Figure 2: Illustrations of: (*left*) the framework DRL with HyAR; and (*right*) overall workflow of hybrid action representation model, consisting of an embedding table and a conditional VAE.

## 3 HYBRID ACTION REPRESENTATION (HYAR)

As mentioned in previous sections, it is non-trivial for an RL agent to learn with discrete-continuous hybrid action space efficiently due to the heterogeneity and action dependence. Naive solutions by converting the hybrid action space into either a discrete or a continuous action space can result in degenerated performance due to the scalability issue and additional approximation difficulty. Previous efforts concentrate on proposing specific policy structures (Hausknecht & Stone, 2016; Fu et al., 2019) that are feasible to learn hybrid-action policies directly over original hybrid action space. However, these methods fail in providing the three desired properties: scalability, stationarity and action dependence simultaneously (See Tab. 1).

Inspired by recent advances in Representation Learning for RL (Whitney et al., 2020; Chandak et al., 2019), we propose Hybrid Action Representation (HyAR), a novel framework that converts the original hybrid-action policy learning into a continuous policy learning problem among the latent action representation space. The intuition behind HyAR is that discrete action and continuous parameter are heterogeneous in their original representations but they jointly influence the environment; thus we can assume that hybrid actions lie on a homogeneous manifold that is closely related to environmental dynamics semantics. In the following, we introduce an unsupervised approach of constructing a compact and decodable latent representation space to approximate such a manifold.

### 3.1 DEPENDENCE-AWARE ENCODING AND DECODING

A desired latent representation space for hybrid actions should take the dependence between the two heterogeneous components into account. Moreover, we need the representation space to be decodable, i.e., the latent actions selected by a latent policy can be mapped back to the original hybrid actions so as to interact with the environment. To this end, we propose dependence-aware encoding and decoding of hybrid action. The overall workflow is depicted in the right of Fig. 2. We establish an embedding table $E_\zeta \in \mathbb{R}^{K \times d_1}$ with learnable parameter $\zeta$ to represent the $K$ discrete actions, where each row $e_{\zeta,k} = E_\zeta(k)$ (with $k$ being the row index) is a $d_1$-dimensional continuous vector for the discrete action $k$. Then, we use a conditional Variational Auto-Encoder (VAE) (Kingma & Welling, 2014) to construct the latent representation space for the continuous parameters. In specific, for a hybrid action $k, x_k$ and a state $s$, the encoder $q_\phi(z \mid x_k, s, e_{\zeta,k})$ parameterized by $\phi$ takes $s$ and the embedding $e_{\zeta,k}$ as condition, and maps $x_k$ into the latent variable $z \in \mathbb{R}^{d_2}$. With the same condition, the decoder $p_\psi(\tilde{x}_k \mid z, s, e_{\zeta,k})$ parameterized by $\psi$ then reconstructs the continuous parameter $\tilde{x}_k$ from $z$. In principle, the conditional VAE can be trained by maximizing the variational lower bound (Kingma & Welling, 2014).

More concretely, we adopt a Gaussian latent distribution $\mathcal{N}(\mu_x, \sigma_x)$ for $q_\phi(z \mid x_k, s, e_{\zeta,k})$ where $\mu_x, \sigma_x$ are the mean and standard deviation outputted by the encoder. For any latent variable sample $z \sim \mathcal{N}(\mu_x, \sigma_x)$, the decoder decodes it deterministicly, i.e., $\tilde{x}_k = p_\psi(z, s, e_{\zeta,k}) = g_{\psi_1} \circ f_{\psi_0}(z, s, e_{\zeta,k})$, where $f_{\psi_0}$ is a transformation network and $g_{\psi_1}$ is a fully-connected layer for reconstruction. We use $\psi = \cup_{i \in \{0,1,2\}} \psi_i$ to denote the parameters of both the decoder network and prediction network (introduced later in Sec. 3.2) as well as the transformation network they share. With a batch of states and original hybrid actions from buffer $\mathcal{D}$, we train the embedding table $E_\zeta$ and the conditional VAE $q_\phi, p_\psi$ together by minimizing the loss function $L_{\text{VAE}}$ below:

$$L_{\text{VAE}}(\phi, \psi, \zeta) = \mathbb{E}_{s,k,x_k \sim \mathcal{D}, z \sim q_\phi}\left[\|x_k - \tilde{x}_k\|_2^2 + D_{\text{KL}}\big(q_\phi(\cdot \mid x_k, s, e_{\zeta,k})\|\mathcal{N}(0, I)\big)\right], \quad (2)$$

where the first term is the $L_2$-norm square reconstruction error and the second term is the Kullback-Leibler divergence $D_{\text{KL}}$ between the variational posterior of latent representation $z$ and the standard Gaussian prior. Note $\tilde{x}_k$ is differentiable with respect to $\psi, \zeta$ and $\phi$ through reparameterization trick (Kingma & Welling, 2014).

The embedding table and conditional VAE jointly construct a compact and decodable hybrid action representation space ($\in \mathbb{R}^{d_1+d_2}$) for hybrid actions. We highlight that this is often much smaller than the joint action space $\mathbb{R}^{K+\sum_k |\mathcal{X}_k|}$ considered in previous works (e.g., PADDPG, PDQN and HPPO), especially when $K$ or $\sum_k |\mathcal{X}_k|$ is large. In this sense, HyAR is expected to be more scalable when compared in Tab. 1. Moreover, the conditional VAE embeds the dependence of continuous parameter on corresponding discrete action in the latent space; and allows to avoid the redundancy of outputting all continuous parameters at any time (i.e., $\mathbb{R}^{\sum_k |\mathcal{X}_k|}$). This resembles the conditional structure adopted by HHQN (Fu et al., 2019) while HyAR is free of the non-stationary issue thanks to learning a single policy in the hybrid representation space.

For any latent variables $e \in \mathbb{R}^{d_1}$ and $z_x \in \mathbb{R}^{d_2}$, they can be decoded into hybrid action $k, x_k$ conveniently by nearest-neighbor lookup of the embedding table along with the VAE decoder. Formally, we summarize the encoding and decoding process below:

$$
\begin{aligned}
&\textbf{Encoding:} \ \ e_{\zeta,k} = E_\zeta(k), \ \ z_x \sim q_\phi(\cdot \mid x_k, s, e_{\zeta,k}) && \text{for } s, k, x_k \\
&\textbf{Decoding:} \ \ k = g_E(e) = \arg\min_{k'\in\mathcal{K}} \|e_{\zeta,k'} - e\|_2, \ \ x_k = p_\psi(z_x, s, e_{\zeta,k}) && \text{for } s, e, z_x
\end{aligned}
\tag{3}
$$

### 3.2 Dynamics Predictive Representation

In the above, we introduce how to construct a compact and decodable latent representation space for original hybrid actions. However, the representation space learned by pure reconstruction of VAE may be pathological in the sense that it is not discriminative to how hybrid actions have different influence on the environment, similarly studied in (Grosnit et al., 2021). Therefore, such a representation space may be ineffective when involved in the learning of a RL policy and value functions, as these functions highly depends on the knowledge of environmental dynamics. To this end, we make full use of environmental dynamics and propose an unsupervised learning loss based on state dynamics prediction to further refine the hybrid action representation.

Intuitively, the dynamics predictive representation learned is semantically smooth. In other words, hybrid action representations that are closer in the space reflects similar influence on environmental dynamics of their corresponding original hybrid actions. Therefore, in principle such a representation space can be superior in the approximation and generalization of RL policy and value functions, than that learned purely from VAE reconstruction. The benefits of dynamics predictive representation are also demonstrated in (Whitney et al., 2020) (Schwarzer et al., 2020).

As shown in the right of Fig. 2, HyAR adopts a subnetwork $h_{\psi_2}$ that is cascaded after the transformation network $f_{\phi_0}$ of the conditional VAE decoder (called *cascaded structure* or *cascaded head* below) to produce the prediction of the state residual of transition dynamics. For any transition sample $(s, k, x_k, s')$, the state residual is denoted by $\delta_{s,s'} = s' - s$. With some abuse of notation (i.e., $p_\psi = h_{\psi_2} \circ f_{\psi_0}$ here), the prediction $\tilde{\delta}_{s,s'}$ is produced as follows, which completes Eq. 3:

$$
\textbf{Prediction:} \ \ \tilde{\delta}_{s,s'} = p_\psi(z_x, s, e_{\zeta,k}) \quad \text{for } s, e, z_x
\tag{4}
$$

Then we minimize the $L_2$-norm square prediction error:

$$
L_{\text{Dyn}}(\phi, \psi, \zeta) = \mathbb{E}_{s,k,x_k,s'} \left[ \|\tilde{\delta}_{s,s'} - \delta_{s,s'}\|_2^2 \right].
\tag{5}
$$

Our structure choice of cascaded prediction head is inspired by (Azabou et al., 2021). The reason behind this is that dynamics prediction could be more complex than continuous action reconstruction, thus usual parallel heads for both reconstruction and the state residual prediction followed by the same latent features may have interference in optimizing individual objectives and hinder the learning of the shared representation.

So far, we derive the ultimate training loss for hybrid action representation as follows:

$$
L_{\text{HyAR}}(\phi, \psi, \zeta) = L_{\text{VAE}}(\phi, \psi, \zeta) + \beta L_{\text{Dyn}}(\phi, \psi, \zeta),
\tag{6}
$$

where $\beta$ is a hyper-parameter that weights the dynamics predictive representation loss. Note that the ultimate loss depends on reward-agnostic data of environmental dynamics, which is dense and usually more convenient to obtain (Stooke et al., 2021; Yarats et al., 2021; Erraqabi et al., 2021).

---

**Algorithm 1:** HyAR-TD3

---

1   Initialize actor $\pi_\omega$ and critic networks $Q_{\theta_1}, Q_{\theta_2}$ with random parameters $\omega, \theta_1, \theta_2$
2   Initialize discrete action embedding table $E_\zeta$ and conditional VAE $q_\phi, p_\psi$ with random parameters $\zeta, \phi, \psi$
3   Prepare replay buffer $\mathcal{D}$
4   **repeat** Stage ❶
5     |   Update $\zeta$ and $\phi, \psi$ using samples in $\mathcal{D}$         ▷ see Eq. 6
6   **until** *reaching maximum warm-up training times*;
7   **repeat** Stage ❷
8     |   **for** $t \leftarrow 1$ *to* $T$ **do**
9     |     // select latent actions in representation space
10   |     $e, z_x = \pi_\omega(s) + \epsilon_\mathrm{e}$, with $\epsilon_\mathrm{e} \sim \mathcal{N}(0, \sigma)$
11   |     // decode into original hybrid actions
12   |     $k = g_E(e), x_k = p_\psi(z_x, s, e_{\zeta,k})$         ▷ see Eq. 3
13   |     Execute $(k, x_k)$, observe $r_t$ and new state $s'$
14   |     Store $\{s, k, x_k, e, z_x, r, s'\}$ in $\mathcal{D}$
15   |     Sample a mini-batch of $N$ experience from $\mathcal{D}$
16   |     Update $Q_{\theta_1}, Q_{\theta_2}$         ▷ see Eq. 7
17   |     Update $\pi_\omega$ with policy gradient         ▷ see Eq. 8
18   |   **repeat**
19   |     Update $\zeta$ and $\phi, \psi$ using samples in $\mathcal{D}$         ▷ see Eq. 6
20   |   **until** *reaching maximum representation training times*;
21   **until** *reaching maximum total environment steps*;

---

## 4   DRL WITH HYBRID ACTION REPRESENTATION

In previous section, we introduce the construction of a compact, decodable and semantically smooth hybrid action representation space. As the conceptual overview in Fig. 1, the next thing is to learn a latent RL policy in the representation space. In principle, our framework is algorithm-agnostic and any RL algorithms for continuous control can be used for implementation. In this paper, we adopt model-free DRL algorithm TD3 (Fujimoto et al., 2018) for demonstration. Though there remains the chance to build a world model based on hybrid action representation, we leave the study on model-based RL with HyAR for future work.

TD3 is popular deterministic-policy Actor-Critic algorithm which is widely demonstrated to be effective in continuous control. As illustrated in the left of Fig. 2, with the learned hybrid action representation space, the actor network parameterizes a latent policy $\pi_\omega$ with parameter $\omega$ that outputs the latent action vector, i.e., $e, z_x = \pi_\omega(s)$ where $e \in \mathbb{R}^{d_1}, z_x \in \mathbb{R}^{d_2}$. The latent action can be decoded according to Eq. 3 and obtain the corresponding hybrid action $k, x_k$. The double critic networks $Q_{\theta_1}, Q_{\theta_2}$ take as input the latent action to approximate hybrid-action value function $Q^{\pi_\omega}$, i.e., $Q_{\theta_{i=1,2}}(s, e, z_x) \approx Q^{\pi_\omega}(s, k, x_k)$. With a buffer of collected transition sample $(s, e, z_x, r, s')$, the critics are trained by Clipped Double Q-Learning, with the loss function below for $i = 1, 2$:

$$L_{\mathrm{CDQ}}(\theta_i) = \mathbb{E}_{s,e,z_x,r,s'} \left[ (y - Q_{\theta_i}(s, e, z_x))^2 \right], \quad \text{where} \ \ y = r + \gamma \min_{j=1,2} Q_{\bar{\theta}_j}\left(s', \pi_{\bar{\omega}}(s')\right), \quad (7)$$

where $\bar{\theta}_{j=1,2}, \bar{\omega}$ are the target network parameters. The actor (latent policy) is updated with Deterministic Policy Gradient (Silver et al., 2014) as follows:

$$\nabla_\omega J(\omega) = \mathbb{E}_s \left[ \nabla_{\pi_\omega(s)} Q_{\theta_1}(s, \pi_\omega(s)) \nabla_\omega \pi_\omega(s) \right]. \quad (8)$$

Algorithm 1 describes the pseudo-code of HyAR-TD3, containing two major stages: ❶ *warm-up stage* and ❷ *training stage*. In the warm-up stage, the hybrid action representation models are pre-trained using a prepared replay buffer $\mathcal{D}$ (line 4-6). The parameters the embedding table and conditional VAE is updated by minimizing the VAE and dynamics prediction loss. Note that the proposed algorithm has no requirement on how the buffer $\mathcal{D}$ is prepared and here we simply use a random policy for the environment interaction and data generation by default. In the learning stage, given a environment state, the latent policy outputs a latent action perturbed by a Gaussian exploration noise, with some abuse of notions $e, z_x$ (line 10). The latent action is decoded into original hybrid action so as to interact with the environment, after which the collected transition sample is stored in the replay buffer (line 12-14). Then, the latent policy learning is preformed using the data sampled from $\mathcal{D}$ (line 15-17). It is worth noting that the action representation model

is updated concurrently in the training stage to make continual adjustment to the change of data distribution (line 19-21).

One significant distinction of DRL with HyAR described above compared with conventional DRL is that, the hybrid action representation space is learned from finite samples that are drawn from a moving data distribution. The induced *unreliability* and *shift* of learned representations can severely cripple the performance of learned latent policy if they are not carefully handled. Hence, we propose two mechanisms to deal with the above two considerations as detailed below.

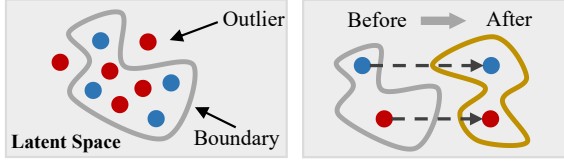

(a) Representation unreliability     (b) Representation shift

Figure 3: Illustrations of representation unreliability and representation shift. Dots denote the hybrid action representations selected by policy (red) and known by encoder (blue). Gray lines form the areas, among which representations can be well decoded and estimated.

**Latent Space Constraint (LSC)** As the latent representation space is constructed by finite hybrid action samples, some areas in the latent space can be highly unreliable in decoding as well as $Q$-value estimation. Similar evidences are also founded in (Zhou et al., 2020; Notin et al., 2021). In Fig. 3(a), the latent action representations inside the boundary can be well decoded and estimated the values, while the outliers cannot. Once the latent policy outputs outliers, which can be common in the early learning stage, the unreliability can quickly deteriorate the policy and lead to bad results. Therefore, we propose to constrain the action representation space of the latent policy inside a reasonable area adaptively. In specific, we re-scale each dimension of the output of latent policy (i.e., $[-1, 1]^{d_1+d_2}$ by tanh activation) to a bounded range $[b_{\text{lower}}, b_{\text{upper}}]$. For a number of $s, k, x_k$ collected previously, the bounds $b_{\text{lower}}, b_{\text{upper}}$ are obtained by calculating the $c$-percentage central range where $c \in [0, 100]$. We empirically demonstrate the importance of LSC. See more details in Appendix A & C.3.

**Representation Shift Correction (RSC)** As in Algorithm 1, the hybrid action representation space is continuously optimized along with RL process. Thus, the representation distribution of original hybrid actions in the latent space can shift after a certain learning interval (Igl et al., 2020). Fig. 3(b) illustrates the shift (denoted by different shapes). This negatively influences the value function learning since the outdated latent action representation no longer reflects the same transition at present. To handle this, we propose a representation relabeling mechanism. In specific, for each mini-batch training in Eq.7, we check the semantic validity of hybrid action representations in current representation space and relabel the invalid ones with the latest representations. In this way, the policy learning is always performed on latest representations, so that the issue of representation shift can be alleviated. Empirically evaluations demonstrate the superiority of relabeling techniques in achieving a better performance with a lower variance. See more details in Appendix A & C.3.

## 5 EXPERIMENTS

We evaluate HyAR in various hybrid action environments against representative prior algorithms. Then, a detailed ablation study is conducted to verify the contribution of each component in HyAR. Moreover, we provide visual analysis for better understandings of HyAR.

### 5.1 EXPERIMENT SETUPS

**Benchmarks** Fig. 4 visualizes the evaluation benchmarks, including the *Platform* and *Goal* from (Masson et al., 2016), *Catch Point* from (Fan et al., 2019), and a newly designed *Hard Move* specific to the evaluation in larger hybrid action space. We also build a complex version of Goal, called *Hard Goal*. All benchmarks have hybrid actions and require the agent to select reasonable actions to complete the task. See complete description of benchmarks in Appendix B.1.

**Baselines** Four state-of-the-art approaches are selected as baselines: HPPO (Fan et al., 2019), PDQN (Xiong et al., 2018), PADDPG (Hausknecht & Stone, 2016), HHQN (Fu et al., 2019). In addition, for a comprehensive study, we extend the baselines which consists of DDPG to their TD3 variants, denoted by PDQN-TD3, PATD3, HHQN-TD3. Last, we use HyAR-DDPG and HyAR-

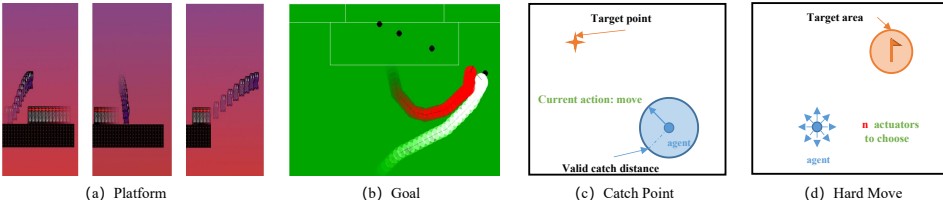

| (a) Platform | (b) Goal | (c) Catch Point | (d) Hard Move |

Figure 4: Benchmarks with discrete-continuous actions: (a) the agent selects a discrete action (run, hop, leap) and the corresponding continuous parameter (horizontal displacement) to reach the goal; (b) The agent selects a discrete strategy (move, shoot) and the continuous 2-D coordinate to score; (c) The agent selects a discrete action (move, catch) and the continuous parameter (direction) to grab the target point; (d) The agent has $n$ equally spaced actuators. It can choose whether each actuator should be on or off (thus $2^n$ combination in total) and determine the corresponding continuous parameter for each actuator (moving distance) to reach the target area.

| ENV | HPPO | PADDPG | PDQN | HHQN | HyAR-DDPG | PATD3 | PDQN-TD3 | HHQN-TD3 | HyAR-TD3 |
|---|---|---|---|---|---|---|---|---|---|
| | **PPO-based** | | **DDPG-based** | | | | **TD3-based** | | |
| Goal | $0.0 \pm 0.0$ | $0.05 \pm 0.10$ | $0.70 \pm 0.07$ | $0.0\pm0.0$ | $0.53\pm0.02$ | $0.0\pm0.0$ | $0.71\pm0.10$ | $0.0\pm0.0$ | **$0.78\pm0.03$** |
| Hard Goal | $0.0 \pm 0.0$ | $0.0 \pm 0.0$ | $0.0 \pm 0.0$ | $0.0\pm0.0$ | $0.30\pm0.08$ | $0.44\pm0.05$ | $0.06\pm0.07$ | $0.01\pm0.01$ | **$0.60\pm0.07$** |
| Platform | $0.80 \pm 0.02$ | $0.36 \pm 0.06$ | $0.93 \pm 0.05$ | $0.46\pm0.25$ | $0.87\pm0.06$ | $0.94\pm0.10$ | $0.93\pm0.03$ | $0.62\pm0.23$ | **$0.98\pm0.01$** |
| Catch Point | $0.69 \pm 0.09$ | $0.82 \pm 0.06$ | $0.77 \pm 0.07$ | $0.31\pm0.06$ | $0.89\pm0.01$ | $0.82\pm0.10$ | $0.89\pm0.07$ | $0.27\pm0.05$ | **$0.90\pm0.03$** |
| Hard Move (n = 4) | $0.09 \pm 0.02$ | $0.03 \pm 0.01$ | $0.69 \pm 0.07$ | $0.39\pm0.14$ | $0.91\pm0.03$ | $0.66\pm0.13$ | $0.85\pm0.10$ | $0.52\pm0.17$ | **$0.93\pm0.02$** |
| Hard Move (n = 6) | $0.05 \pm 0.01$ | $0.04 \pm 0.01$ | $0.41 \pm 0.05$ | $0.32\pm0.17$ | $0.91\pm0.04$ | $0.04\pm0.02$ | $0.74\pm0.08$ | $0.29\pm0.13$ | **$0.92\pm0.04$** |
| Hard Move (n = 8) | $0.04 \pm 0.01$ | $0.06 \pm 0.03$ | $0.04 \pm 0.01$ | $0.05\pm0.02$ | $0.85\pm0.06$ | $0.06\pm0.02$ | $0.05\pm0.01$ | $0.05\pm0.02$ | **$0.89\pm0.03$** |
| Hard Move (n = 10) | $0.05 \pm 0.01$ | $0.04 \pm 0.01$ | $0.06 \pm 0.02$ | $0.04\pm0.01$ | **$0.82\pm0.06$** | $0.07\pm0.02$ | $0.05\pm0.02$ | $0.05\pm0.02$ | $0.75\pm0.05$ |

Table 2: Comparisons of the baselines regarding the average performance at the end of training process with the corresponding standard deviation over 5 runs. Values in bold indicate the best results in each environment.

TD3 to denote our implementations of DRL with HyAR based on DDPG and TD3. For a fair comparison, the network architecture (i.e., DDPG and TD3) used in associated baselines are the same. For all experiments, we give each baseline the same training budget. For our algorithms, we use a random strategy to interact with the environment for 5000 episodes during the warm-up stage. For each experiment, we run 5 trials and report the average results. Complete details of setups are provided in Appendix B.

## 5.2 PERFORMANCE EVALUATION

To conduct a comprehensive comparison, all baselines implemented based on either DDPG or TD3 are reported. To counteract implementation bias, codes of PADDPG, PDQN, and HHQN are directly adopted from prior works. Comparisons in terms of the averaged results are summarized in Tab. 2, where bold numbers indicate the best result. Overall, we have the following findings. HyAR-TD3 and HyAR-DDPG show the better results and lower variance than the others. Moreover, the advantage of HyAR is more obvious in environments in larger hybrid action space (e.g., Hard Goal & Hard Move). Taking Hard Move for example, as the action space grows exponentially, the performance of HyAR is steady and barely degrades, while the others deteriorate rapidly. Similar results can be found in Goal and Hard Goal environments. This is due to the superiority of HyAR of utilizing the hybrid action representation space, among which the latent policy can be learned based on compact semantics. These results not only reveal the effectiveness of HyAR in achieving better performance, but also the scalability and generalization.

In almost all environments, HyAR outperforms other baselines for both the DDPG-based and TD3-based cases. The exceptions are in Goal and Platform environments, where PDQN performs slightly better than HyAR-DDPG. We hypothesize that this is because the hybrid action space of these two environments is relatively small. For such environments, the learned latent action space could be sparse and noisy, which in turn degrades the performance. One evidence is that the conservative (underestimation) nature in TD3 could compensate and alleviates this issue, achieving significant improvements (HyAR-TD3 v.s. HyAR-DDPG). Fig. 5 renders the learning curves, where HyAR-TD3 outperforms other baselines in both the final performance and learning speed across all environments. Similar results are observed in DDPG-based comparisons and can be found in Appendix C.1. In addition, HyAR-TD3 shows good generalization across environments, while the others more or

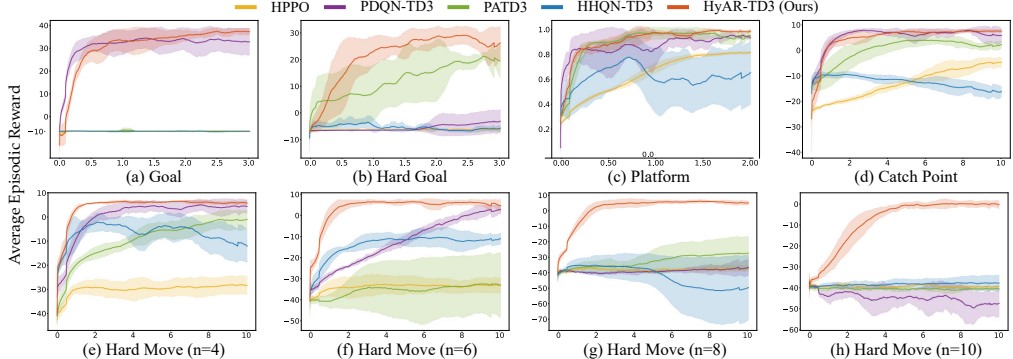

Figure 5: Comparisons of algorithms in different environments. The x- and y-axis denote the environment steps ($\times 10^5$) and average episode reward over recent 100 episodes. The curve and shade denote the mean and a standard deviation over 5 runs.

less fail in some environments (e.g., HPPO, PATD3, and HHQN-TD3 fail in Fig. 5(a) and PDQN-TD3 fails in Fig. 5(b)). Moreover, when environments become complex (Fig. 5(e-h)), HyAR-TD3 still achieves steady and better performance, particularly demonstrating the effectiveness of HyAR in high-dimensional hybrid action space.

## 5.3 ABLATION STUDY AND VISUAL ANALYSIS

We further evaluate the contribution of the major components in HyAR: the two mechanisms for latent policy learning, i.e., latent space constraint (LSC) and representation shift correction (RSC), and the dynamics predictive representation loss. We briefly conclude our results as follows. For LSC, properly constraining the output space of the latent policy is critical to performance; otherwise, both loose and conservative constraints dramatically lead to performance degradation. RSC and dynamics predictive representation loss show similar efficacy: they improve both learning speed and convergence results, additionally with a lower variance. Such superiority is more significant in the environment when hybrid actions are more semantically different (e.g., Goal). We also conduct ablation studies on other factors along with hyperparameter analysis. See complete details and ablation results in Appendix C.2 & C.3.

Finally, we adopt t-SNE (Maaten & Hinton, 2008) to visualize the learned hybrid action representations, i.e., $(e, z_x)$, in a 2D plane. We color each action based on its impact on the environment i.e., $\tilde{\delta}_{s,s'}$. As shown in Fig. 6, we observe that actions with a similar impact on the environment are relatively closer in the latent space. This demonstrates the dynamics predictive representation loss is helpful for deriving dynamics-aware representation for further improving the learning performance, efficacy, and stability (see results in Appendix C.2 & C.4)

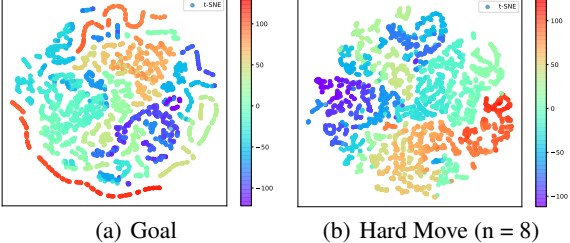

(a) Goal      (b) Hard Move (n = 8)

Figure 6: 2D t-SNE visualizations of learned representation for original hybrid actions, colored by 1D t-SNE of the corresponding environmental impact.

## 6 CONCLUSION

In this paper, we propose Hybrid Action Representation (HyAR) for DRL agents to efficiently learn with discrete-continuous hybrid action space. HyAR use an unsupervised method to derive a compact and decodable representation space for discrete-continuous hybrid actions. HyAR can be easily extended with modern DRL methods to leverage additional advantages. Our experiments demonstrate the superiority of HyAR regarding performance, learning speed and robustness in most hybrid action environment, especially in high-dimensional action spaces.

## ACKNOWLEDGMENTS

The work is supported by the National Science Fund for Distinguished Young Scholars (Grant No.: 62025602), the National Natural Science Foundation of China (Grant Nos.: 11931015, 62106172), the XPLORER PRIZE, and the New Generation of Artificial Intelligence Science and Technology Major Project of Tianjin (Grant No.: 19ZXZNGX00010).

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

# A   DETAILED FOR LATENT SPACE CONSTRAINT (LSC) AND REPRESENTATION SHIFT CORRECTION(RSC)

**Latent Space Constraint (LSC)**   As we can see in Fig. 3(a), the latent action representations inside the boundary can be well decoded and estimated the values, while the outliers cannot. Therefore, the most critical problem for latent space constraint (LSC) is to find a reasonable latent space boundary. Simply re-scale policy's outputs in a fixed bounded area $[-b, b]$ could lose some important information and make the latent space unstable (Zhou et al., 2020; Notin et al., 2021) . We propose a mechanism to constrain the action representation space of the latent policy inside a reasonable area adaptively. In specific, we re-scale each dimension of the output of latent policy (i.e., $[-1, 1]^{d_1+d_2}$ by tanh activation) to a bounded range $[b_{\text{lower}}, b_{\text{upper}}]$. At intervals (actually concurrent with the updates of the hybrid action representation models), we first sample $M$ transitions $s, k, x_k$ from buffer, then we obtain the corresponding latent action representations with current representation models. In this way, we will get $M$ different latent variable values in each dimension. We sort the latent variable of each dimension and calculate the $c$-percentage central range, i.e., let the $\frac{c}{2}$ quantile and $1 - \frac{c}{2}$ quantile of the range to be $b_{\text{lower}}$ and $b_{\text{upper}}$ of the current latent variable. We called $c$ as latent select range where $c \in [0, 100]$. With the decrease of $c$, the constrained latent action representation space becomes smaller. The experiment on the value of latent select range $c$ is in Appendix C.3.

**Representation Shift Correction (RSC)**   Since the hybrid action representation space is continuously optimized along with the RL learning, the representation distribution of original hybrid actions in the latent space can shift after a certain learning interval (Igl et al., 2020). Fig. 3(b) visualizes the shifting (denoted by different shapes). This negatively influences the value function learning since the outdated latent action representation no longer reflects the same transition at present. To handle this, we propose a representation relabeling mechanism. In specific, we feed the batch of stored original hybrid actions to our representation models to obtain the latest latent representations, for each mini-batch training in Eq.7. For latent discrete action $e$, if it can not be mapped to the corresponding original action $k$ in the latest embedding table, we will relabel $e$ through looking up the table with stored original discrete action $k$, i.e., $e \leftarrow e_{\zeta,k} + \mathcal{N}(0, 0.1)$. The purpose of adding noise $\mathcal{N}(0, 0.1)$ is to ensure the diversity of the relabeled action representations. For latent continuous action $z_x$, we first obtain $\tilde{\delta}_{s,s'}$ through the latest decoder $p_\psi(z_x, s, e_{\zeta,k})$. Then we verify if $\|\tilde{\delta}_{s,s'} - \delta_{s,s'}\|_2^2 > \delta_0$ (threshold value $\delta_0$ is set to be $4 * \hat{L}_{\text{Dyn}}$, where $\hat{L}_{\text{Dyn}}$ is the moving empirical loss), i.e., the case that indicates that the historical representations has no longer semantically consistent (with respect to environmental dynamics) under current representation models. Then $z_x$ will be relabeled by the latest latent representations $z \sim q_\phi(\cdot \mid x_k, s, e_{\zeta,k})$. In this way, the policy learning is always performed on latest representations, so that the issue of representation distribution shift can be effectively alleviated. The experiment on relabeling techniques is in Appendix C.3.

**Limitation and Potential Improvements of LSC and RSC**   The ideas of LSC and RSC are general. In our work, both LSC and RSC are easy to implement and empirically demonstrated to be effective. We also consider that there remains some room to further improve these two mechanisms in some more challenging cases. For LSC, one potential drawback may be that the central range constraint on per dimension adopted in our work, is suboptimal to the case if the distribution of learned action representation of original hybrid actions is multi-modal. In such cases, the central range constraint may be less effective and can be further improved. For RSC, it requires additional memory and computation cost. However, in our experiments, such memory and computation cost is relatively negligible. The dimensionality of each original hybrid action sample is relatively small compared to the state sample stored; the additional wall-clock time cost of HyAR (with RSC) to HyAR without RSC is about 6% (tested in *Goal*).

# B   EXPERIMENTAL DETAILS

## B.1   SETUPS

Our codes are implemented with Python 3.7.9 and Torch 1.7.1. All experiments were run on a single NVIDIA GeForce GTX 2080Ti GPU. Each single training trial ranges from 4 hours to 10 hours, depending on the algorithms and environments. For more details of our code can refer to the HyAR.zip in the supplementary results.

| Layer | Actor Network ($\pi(s)$) | Critic Network ($Q(s,a)$ or $V(s)$) |
|---|---|---|
| Fully Connected | (state dim, 256) | (state dim + $\mathbb{R}^{K+\sum_k |\mathcal{X}_k|}$, 256) or (state dim + $\mathbb{R}^{\sum_k |\mathcal{X}_k|}$, 256) or (state dim, 256) |
| Activation | ReLU | ReLU |
| Fully Connected | (256, 256) | (256, 256) |
| Activation | ReLU | ReLU |
| Fully Connected | (256, $\mathbb{R}^K$) and (256, $\mathbb{R}^{\sum_k |\mathcal{X}_k|}$) or (256, $\mathbb{R}^{\sum_k |\mathcal{X}_k|}$) | (256, 1) or (256, $\mathbb{R}^K$) |
| Activation | Tanh | None |

Table 3: Network structures for the actor network and the critic network ($Q$-network or $V$-network).

**Benchmark Environments** We conduct our experiments on several hybrid action environments and detailed experiment description is below.

- ***Platform*** (Masson et al., 2016): The agent need to reach the final goal while avoiding the enemy or falling into the gap. The agent need to select the discrete action (run, hop, leap) and determine the corresponding continuous action (horizontal displacement) simultaneously to complete the task. The horizon of an episode is 20.

- ***Goal*** (Masson et al., 2016): The agent shoots the ball into the gate to win. Three types of hybrid actions are available to the agent including *kick-to(x,y), shoot-goal-left(h), shoot-goal-right(h)*. The continuous action parameters position (x, y) and position (h) along the goal line are quit different. Furthermore, We built a complex version of the goal environment, called ***Hard Goal***. We redefined the shot-goal action and split it into ten parameterized actions by dividing the goal line equidistantly. The continuous action parameters of each shot action will be mapped to a region in the goal line. The horizon of an episode is 50.

- ***Catch Point*** (Fan et al., 2019): The agent should catch the target point (orange) in limited opportunity (10 chances). There are two hybrid actions *move* and *catch*. Move is parameterized by a continuous action value which is a directional variable and catch is to try to catch the target point. The horizon of an episode is 20.

- ***Hard Move (designed by us)***: The agent needs to control $n$ equally spaced actuators to reach target area (orange). Agent can choose whether each actuator should be on or off. Thus, the size of the action set is exponential in the number of actuators that is $\mathbf{2^n}$. Each actuator controls the moving distance in its own direction. $n$ controls the scale of the action space. As n increases, the dimension of the action will increase. The horizon of an episode is 25.

## B.2 NETWORK STRUCTURE

Our PATD3 is implemented with reference to github.com/sfujim/TD3 (TD3 source-code). PADDPG and PDQN are implemented with reference to https://github.com/cycraig/MP-DQN. For a fair comparison, all the baseline methods have the same network structure (except for the specific components to each algorithm) as our HyAR-TD3 implementation. For PDQN, PADDPG, we introduce a Passthrough Layer (Masson et al., 2016) to the actor networks to initialise their action-parameter policies to the same linear combination of state variables. HPPO paper does not provide open source-code and thus we implemented it by ourselves according to the guidance provided in their paper. For HPPO, the discrete actor and continuous actor do not share parameters (better than share parameters in our experiments).

As shown in Tab. 3, we use a two-layer feed-forward neural network of 256 and 256 hidden units with ReLU activation (except for the output layer) for the actor network for all algorithms. For PADDPG, PDQN and HHQN, the critic denotes the $Q$-network. For HPPO, the critic denotes the $V$-network. Some algorithms (PATD3, PADDPG, HHQN) output two heads at the last layer of the actor network, one for discrete action and another for continuous action parameters.

The structure of HyAR is shown in Tab. 4. We introduced element-wise product operation (Tang et al., 2021) and cascaded head structure (Azabou et al., 2021) to our HyAR model. More details about their effects are in Appendix C.3.

| Model Component | Layer (Name) | Structure |
|---|---|---|
| Discrete Action Embedding Table $E_\zeta$ | Parameterized Table | $(\mathbb{R}^{d_1}, \mathbb{R}^K)$ |
| Conditional Encoder Network $q_\phi\left(z \mid x_k, s, e_{\zeta,k}\right)$ | Fully Connected (encoding) | $(\mathbb{R}^{\mathcal{X}_k}, 256)$ |
| | Fully Connected (condition) | (state dim + $\mathbb{R}^{d_1}$, 256) |
| | Element-wise Product | ReLU (encoding) · ReLU(condition) |
| | Fully Connected | (256, 256) |
| | Activation | ReLU |
| | Fully Connected (mean) | (256, $\mathbb{R}^{d_2}$) |
| | Activation | None |
| | Fully Connected (log_std) | (256, $\mathbb{R}^{d_2}$) |
| | Activation | None |
| Conditional Decoder & Prediction Network $p_\psi(x_k, \tilde{\delta}_{s,s'} \mid z_k, s, e_{\zeta,k})$ | Fully Connected (latent) | $(\mathbb{R}^{d_2}, 256)$ |
| | Fully Connected (condition) | (state dim + $\mathbb{R}^{d_1}$, 256) |
| | Element-wise Product | ReLU(decoding) · ReLU(condition) |
| | Fully Connected | (256, 256) |
| | Activation | ReLU |
| | Fully Connected (reconstruction) | (256, $\mathbb{R}^{\mathcal{X}_k}$) |
| | Activation | None |
| | Fully Connected | (256, 256) |
| | Activation | ReLU |
| | Fully Connected (prediction) | (256, state dim) |
| | Activation | None |

Table 4: Network structures for the hybrid action representation (HyAR) including, the discrete action embedding table and the conditional VAE.

## B.3 HYPERPARAMETER

For all our experiments, we use the raw state and reward from the environment and no normalization or scaling are used. No regularization is used for the actor and the critic in all algorithms. An exploration noise sampled from $N(0, 0.1)$ (Fujimoto et al., 2018) is added to all baseline methods when select action. The discounted factor is 0.99 and we use Adam Optimizer (Kingma & Ba, 2015) for all algorithms. Tab. 5 shows the common hyperparamters of algorithms used in all our experiments.

| Hyperparameter | HPPO | PADDPG | PDQN | HHQN | PATD3 | PDQN-TD3 | HHQN-TD3 | HyAR-DDPG | HyAR-TD3 |
|---|---|---|---|---|---|---|---|---|---|
| Actor Learning Rate | $1 \cdot 10^{-4}$ | $1 \cdot 10^{-4}$ | $1 \cdot 10^{-4}$ | $1 \cdot 10^{-4}$ | $3 \cdot 10^{-4}$ | $3 \cdot 10^{-4}$ | $3 \cdot 10^{-4}$ | $1 \cdot 10^{-4}$ | $3 \cdot 10^{-4}$ |
| Critic Learning Rate | $1 \cdot 10^{-3}$ | $1 \cdot 10^{-3}$ | $1 \cdot 10^{-3}$ | $1 \cdot 10^{-3}$ | $3 \cdot 10^{-4}$ | $3 \cdot 10^{-4}$ | $3 \cdot 10^{-4}$ | $1 \cdot 10^{-3}$ | $3 \cdot 10^{-4}$ |
| Representation Model Learning Rate | - | - | - | - | - | - | - | $1 \cdot 10^{-4}$ | $1 \cdot 10^{-4}$ |
| Discount Factor | 0.99 | 0.99 | 0.99 | 0.99 | 0.99 | 0.99 | 0.99 | 0.99 | 0.99 |
| Optimizer | Adam | Adam | Adam | Adam | Adam | Adam | Adam | Adam | Adam |
| Target Actor Update Rate | - | $1 \cdot 10^{-3}$ | $1 \cdot 10^{-3}$ | $1 \cdot 10^{-3}$ | $5 \cdot 10^{-3}$ | $5 \cdot 10^{-3}$ | $5 \cdot 10^{-3}$ | $1 \cdot 10^{-3}$ | $5 \cdot 10^{-3}$ |
| Target Critic Update Rate | - | $1 \cdot 10^{-2}$ | $1 \cdot 10^{-2}$ | $1 \cdot 10^{-2}$ | $5 \cdot 10^{-3}$ | $5 \cdot 10^{-3}$ | $5 \cdot 10^{-3}$ | $5 \cdot 10^{-3}$ | $5 \cdot 10^{-3}$ |
| Exploration Policy | $\mathcal{N}(0, 0.1)$ | $\mathcal{N}(0, 0.1)$ | $\mathcal{N}(0, 0.1)$ | $\mathcal{N}(0, 0.1)$ | $\mathcal{N}(0, 0.1)$ | $\mathcal{N}(0, 0.1)$ | $\mathcal{N}(0, 0.1)$ | $\mathcal{N}(0, 0.1)$ | $\mathcal{N}(0, 0.1)$ |
| Batch Size | 128 | 128 | 128 | 128 | 128 | 128 | 128 | 128 | 128 |
| Buffer Size | $10^5$ | $10^5$ | $10^5$ | $10^5$ | $10^5$ | $10^5$ | $10^5$ | $10^5$ | $10^5$ |
| Actor Epoch | 2 | - | - | - | - | - | - | - | - |
| Critic Epoch | 10 | - | - | - | - | - | - | - | - |

Table 5: A comparison of common hyperparameter choices of algorithms. We use '-' to denote the 'not applicable' situation.

## B.4 ADDITIONAL IMPLEMENTATION DETAILS

**Training setup:** For HPPO, the actor network and the critic network are updated every 2 and 10 episodes respectively for all environment. The clip range of HPPO algorithm is set to 0.2 and we use GAE (Schulman et al., 2016) for stable policy gradient. For DDPG-based, the actor network and the critic network is updated every 1 environment step. For TD3-based, the critic network is updated every 1 environment step and the actor network is updated every 2 environment step.

The discrete action embedding table is initialized randomly by drawing each dimension from the uniform distribution $U(-1, 1)$ before representation pre-training. The latent action dim (discrete or

---

**Algorithm 2:** HyAR-DDPG

---

1 Initialize actor $\pi_\omega$ and critic networks $Q_\theta$ with random parameters $\omega, \theta$, and the corresponding target network parameters $\bar{\omega}, \bar{\theta}$
2 Initialize discrete action embedding table $E_\zeta$ and conditional VAE $q_\phi, p_\psi$ with random parameters $\zeta, \phi, \psi$
3 Prepare replay buffer $\mathcal{D}$
4 **repeat** Stage ❶
5    | Update $\zeta$ and $\phi, \psi$ using samples in $\mathcal{D}$             ▷ see Eq. 6
6 **until** *reaching maximum warm-up training times*;
7 **repeat** Stage ❷
8    | **for** $t \leftarrow 1$ *to* $T$ **do**
9         | // select latent actions in representation space
10         | $e, z_x = \pi_\omega(s) + \epsilon_{\mathrm{e}}$, with $\epsilon_{\mathrm{e}} \sim \mathcal{N}(0, \sigma)$
11         | // decode into original hybrid actions
12         | $k = g_E(e), x_k = p_\psi(z_x, s, e_{\zeta,k})$          ▷ see Eq. 3
13         | Execute $(k, x_k)$, observe $r_t$ and new state $s'$
14         | Store $\{s, k, x_k, e, z_x, r, s'\}$ in $\mathcal{D}$
15         | Sample a mini-batch $B$ of $N$ experience from $\mathcal{D}$
16         | Update critic by minimizing empirical loss $\hat{L}_Q(\theta) = N^{-1} \sum_B (y - Q_\theta(s, e, z_x))^2$, where $y = r + \gamma Q_{\bar\theta}(s', \pi_{\bar\omega}(s'))$
17         | Update actor by the deterministic policy gradient
18         | $\nabla_\omega J(\omega) = N^{-1} \sum_{s \in B} \left[ \nabla_{\pi_\omega(s)} Q_\theta(s, \pi_\omega(s)) \nabla_\omega \pi_\omega(s) \right]$.
19    | **repeat**
20         | Update $\zeta$ and $\phi, \psi$ using samples in $\mathcal{D}$      ▷ see Eq. 6
21    | **until** *reaching maximum representation training times*;
22 **until** *reaching maximum total environment steps*;

---

continuous latent action) default value is 6. We set the KL weight in representation loss $L_{\mathrm{VAE}}$ as 0.5 and dynamics predictive representation loss weight $\beta$ as 10 (default). More details about dynamics predictive representation loss weight are in Appendix C.2.

For the warm-up stage, we run 5000 episodes (please refer to Tab. 7 for the corresponding environment steps in different environments) for experience collection. We then pre-train the hybrid action representation model (discrete action embedding table and conditional VAE) for 5000 batches with batch size 64, after which we start the training of the latent policy. The representation models (the embedding table and conditional VAE) are trained every 10 episodes for 1 batches with batch size 64 for the rest of RL training. See Appendix D for different choices of warm-up training and subsequent training of the hybrid action representation.

### B.5 DDPG-BASED HYAR ALGORITHM

Additionally, we implemented HyAR with DDPG (Lillicrap et al., 2015), called HyAR-DDPG. The pseudo-code of complete algorithm is shown in Algorithm 2. Results of DDPG-based experimental comparisons can be found in Appendix C.1.

## C COMPLETE LEARNING CURVES AND ADDITIONAL EXPERIMENTS

### C.1 LEARNING CURVES FOR DDPG-BASED COMPARISONS

Fig. 7 visualizes the learning curves of DDPG-based comparisons, where HyAR-DDPG outperforms other baselines in both the final performance and learning speed in most environments. Besides the learning speed, HyAR-DDPG also achieves the best generalization as HyAR-TD3 across different environments. When the environments become complex (shown in Fig. 7(e-h)), HyAR-DDPG still achieves steady and better performance than the others, particularly demonstrating the effectiveness and generalization of HyAR in high-dimensional hybrid action spaces.

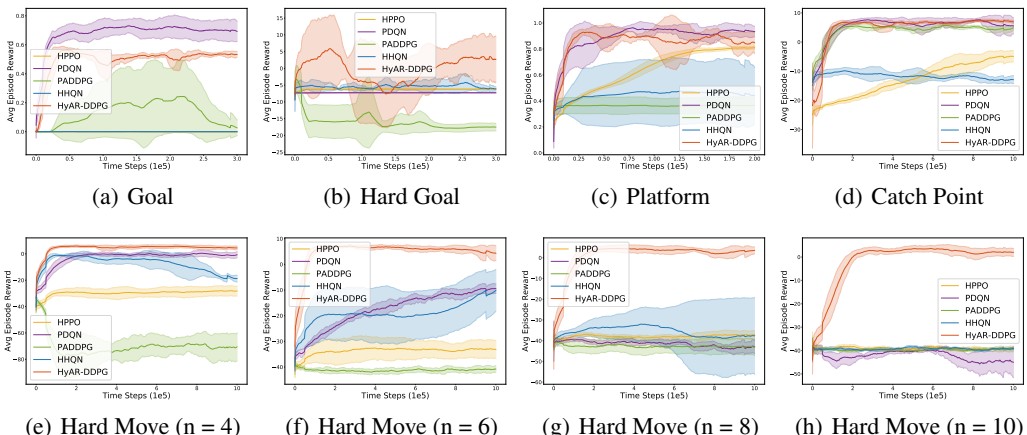

(a) Goal     (b) Hard Goal     (c) Platform     (d) Catch Point

(e) Hard Move (n = 4)     (f) Hard Move (n = 6)     (g) Hard Move (n = 8)     (h) Hard Move (n = 10)

Figure 7: DDPG-based comparisons of related baselines in different environments. The x- and y-axis denote the environment steps ($\times 10^5$) and average episode reward over recent 100 episodes. The results are averaged using 5 runs, while the solid line and shaded represent the mean value and a standard deviation respectively.

## C.2 LEARNING CURVES FOR THE DYNAMICS PREDICTIVE REPRESENTATION

Fig. 8 shows the learning curves of HyAR-TD3 with dynamics predictive representation loss (Fig. 8(a-b)) and the influence of dynamics predictive representation loss weight $\beta$ on algorithm performance (Fig. 8(c)). We can easily find that the representation learned by dynamics predictive representation loss is better than without dynamics predictive representation loss. For the weight $\beta$ of dynamics predictive representation loss, we search the candidate set $\{0.1, 1, 5, 10, 20\}$. The results show that the performance of the algorithm gradually improves with the increase of weight $\beta$, reaches the best when $\beta = 10$ and then goes down as further increase of $\beta$. We can conclude that the dynamics predictive representation loss is helpful for deriving an environment-awareness representation for further improving the learning performance, efficacy, and stability. More experiments on representation visualization are in Appendix C.4.

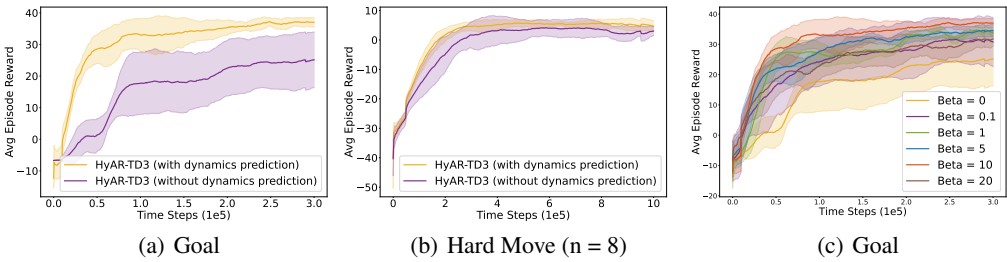

(a) Goal     (b) Hard Move (n = 8)     (c) Goal

Figure 8: Learning curves for variants of dynamics predictive representation in HyAR. $(a)$ and $(b)$ show the comparison between HyAR-TD3 with and without dynamics prediction auxiliary loss; $(c)$ shows the effects of the balancing weight $\beta$ in Eq.6. The results are averaged using 5 runs, while the solid line and shaded represent the mean value and a standard deviation respectively.

## C.3 LEARNING CURVES AND TABLE FOR THE RESULTS IN ABLATION STUDY

As briefly discussed in Sec. 5.3, we conduct detailed ablation and parameter analysis experiments on the key components of the algorithm, including:

- element-wise product (Tang et al., 2021) (v.s. concat) operation;
- cascaded head (Azabou et al., 2021) (v.s. parallel head) structure;

- latent select range $c \in \{80, 90, 96, 100\}$, for the latent space constraint (LSC) mechanism;
- action representation relabeling, corresponding to representation shift correction (RSC);
- latent action dim $d_1 = d_2 \in \{3, 6, 12\}$;

Fig. 9 shows the learning curves of HyAR-TD3 and its variants for ablation studies, corresponding to the results in Tab. 6.

First, we can observe that element-wise product achieves better performance than concatenation (Fig. 9(a,e)). As similarly discovered in (Tang et al., 2021), we hypothesize that the explicit relation between the condition and representation imposed by element wise product forces the conditional VAE to learn more effective hidden features. Second, the significance of cascaded head is demonstrated by its superior performance over parallel head (Fig. 9(a,e)) which means cascaded head can better output two different features. Third, representation relabeling shows an apparent improvement (Fig. 9(b,f)) which show that representation shift leads to data invalidation in the experience buffer which will affect RL training. Fourth, a reasonable latent select range plays an important role in algorithm learning (Fig. 9(c,g)). Only constrain the action representation space of the latent policy inside a reasonable area (both large and small will fail), can the algorithm learn effectively and reliably. These experimental results supports our analysis above.

We also analyse the influence of latent action dim $d_1, d_2$ for RL (Fig. 9(d,h)). In the low-dimensional hybrid action environment, we should choose a moderate value (e.g., 6). While for high-dimensional environment, larger value may be better (e.g., 12). The insight behind is that the proper dimensionality of latent action representation may be comparable (or more compact) to the dimensionality of state (ranging from 4 to 17 dimensions in different environments in our experiments). This is because the latent action representation should reflect the semantics of original hybrid action, i.e., the state residual.

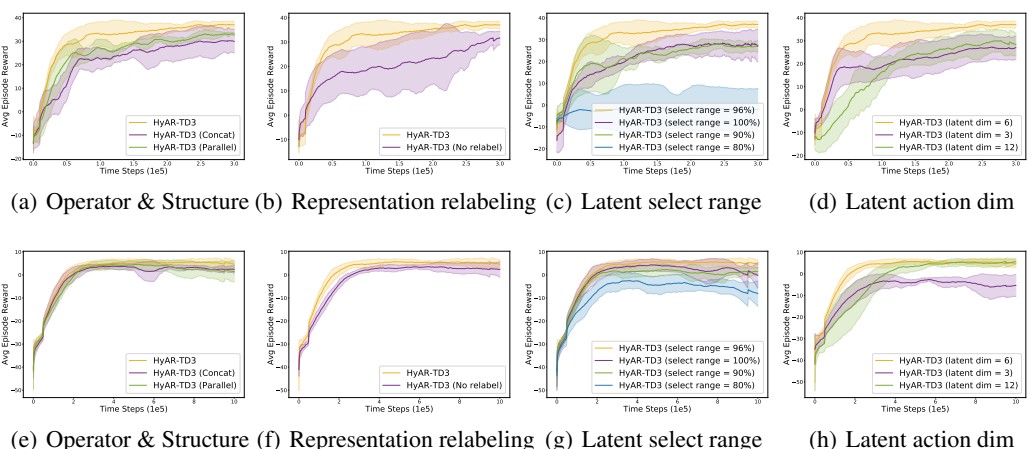

(a) Operator & Structure (b) Representation relabeling (c) Latent select range (d) Latent action dim

(e) Operator & Structure (f) Representation relabeling (g) Latent select range (h) Latent action dim

Figure 9: Learning curves of ablation studies for HyAR (i.e., element-wise + cascaded head + representation relabeling + latent select range = 96% + latent action dim = 6) corresponding to Tab. 6. From top to bottom is Goal and Hard Move ($n = 8$) environment. The shaded region denotes standard deviation of average evaluation over 5 runs.

## C.4 REPRESENTATION VISUAL ANALYSIS

In order to further analyze the hybrid action representation, we visualize the learned hybrid action representations. Fig. 10 and Fig. 11 shows the t-SNE visualization for HyAR in Goal and Hard Move ($n = 8$) environment.

As we can see from Fig. 10, we adopt t-SNE to cluster the latent continuous actions, i.e., $(z_x)$, outputted by the latent policy, and color each action based on latent discrete actions i.e., $(e)$. We can conclude that latent continuous actions can be clustered by latent discrete actions, but there are multiple modes in the global range. Our dependence-aware representation model makes good use of this relationship that the choice of continuous action parameters is depend on discrete actions.

| Operation | | Structure | | | | | | Result | |
| Elem.-Wise Prod. | Concat. | Cascaded | Parallel | Latent Select Range $c$ | Latent Action Dim | Relabeling | Dynamics Predictive | Results (Goal) | Results (Hard Move) |
|---|---|---|---|---|---|---|---|---|---|
| ✓ | | ✓ | | 96% | 6 | ✓ | ✓ | **0.78 ± 0.03** | 0.89 ± 0.03 |
| | ✓ | ✓ | | 96% | 6 | ✓ | ✓ | 0.66± 0.10 | 0.83± 0.04 |
| ✓ | | | ✓ | 96% | 6 | ✓ | ✓ | 0.71 ± 0.04 | 0.80± 0.13 |
| ✓ | | ✓ | | 96% | 6 | | ✓ | 0.66 ± 0.07 | 0.83± 0.08 |
| ✓ | | ✓ | | 100% | 6 | ✓ | ✓ | 0.62 ± 0.11 | 0.78± 0.13 |
| ✓ | | ✓ | | 90% | 6 | ✓ | ✓ | 0.61 ± 0.04 | 0.78± 0.08 |
| ✓ | | ✓ | | 80% | 6 | ✓ | ✓ | 0.08 ± 0.17 | 0.56± 0.12 |
| ✓ | | ✓ | | 96% | 3 | ✓ | ✓ | 0.59 ± 0.09 | 0.58± 0.16 |
| ✓ | | ✓ | | 96% | 12 | ✓ | ✓ | 0.65 ± 0.09 | **0.90 ± 0.04** |
| ✓ | | ✓ | | 96% | 6 | ✓ | | 0.55 ± 0.15 | 0.84± 0.05 |

Table 6: Ablation of our algorithm across each contribution in Goal and Hard Move ($n = 8$). Results are average success rates at the end of training process over 5 runs. ± corresponds to a standard deviation. The corresponding episode reward learning curves are shown in Fig. 9.

For the dynamics predictive representation loss, we adopt t-SNE to cluster the latent actions, i.e., $(e, z_x)$, outputted by the latent policy, and color each action based on its impact on the environment (i.e., $\delta_{s,s'}$). As shown in Fig. 11, we observe that actions with a similar impact on the environment are relatively closer in the latent space. This demonstrates the dynamics predictive representation loss is helpful for deriving an environment-awareness representation for further improving the learning performance, efficacy, and stability.

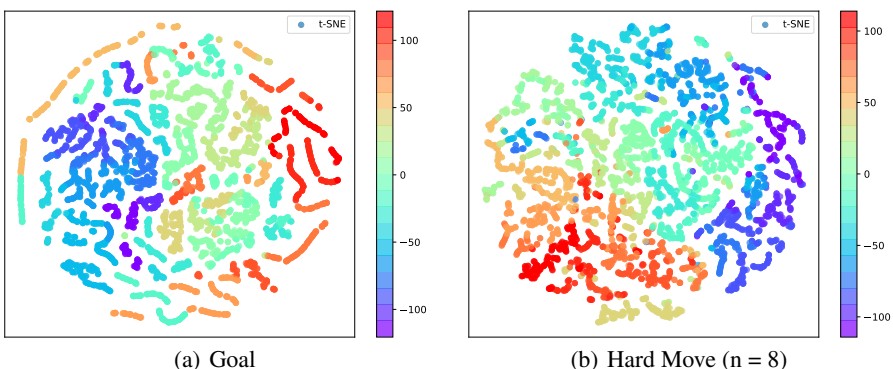

(a) Goal         (b) Hard Move (n = 8)

Figure 10: t-SNE visualization diagram of continuous action embedding $z_x$, color coded by discrete action embedding $e$. The continuous actions related to the same discrete actions are mapped to the similar regions of the representation space.

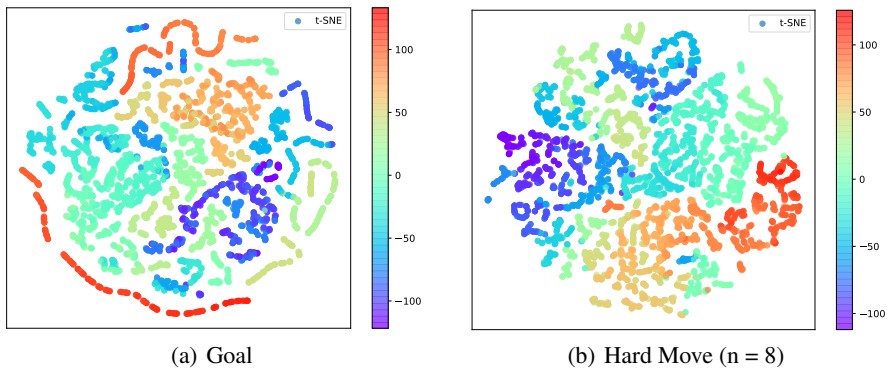

(a) Goal         (b) Hard Move (n = 8)

Figure 11: t-SNE visualization diagram of hybrid action embedding pair $(e, z_x)$, color coded by $\tilde{\delta}_{s,s'}$. The hybrid actions with a similar impact on the environment are relatively closer in the latent space.

# D    ADDITIONAL EXPERIMENTS

In this section, we conduct some additional experimental results for a further study of HyAR from different perspectives:

- We provide the exact number of samples used in the warm-up stage (i.e., stage 1 in Algorithm 1 in each environment in Tab. 7. The numbers of warm-up environment steps are about $5\%-10\%$ of the total environment steps in our original experiments.

- Moreover, we also conduct some experiments to further reduce the number of samples used in the warm-up stage (at most 80% off). See the colored results in Tab. 7. HyAR can achieve comparable performance with $< 3\%$ samples of the total environment steps.

- We also provide Fig. 12 for another view of the learning curves where the number of samples used in the warm-up stage is also counted for HyAR.

- We conduct additional experiments to compare HyAR-TD3 and HyAR-TD3 with fixed hybrid action representation space trained in the warm-up stage in the environments *Platform, Goal, Hard Goal, Catch Point and Hard Move* ($n = 8$). The results are provided in Fig. 13, demonstrating the necessity of subsequent training of the representation trained in the warm-up stage.

- We further conduct experiments to study the effects of different choices of training frequency of subsequent representation training, in the environments *Goal, Hard Goal*. The results are provided in Fig. 13, demonstrating a moderate training frequency works best.

| Environment | Number of Warm-up Env. Steps original | new | Number of Total Env. Steps |
|---|---|---|---|
| Goal | 20000 (0.067\|0.78) | 5000 (0.017\|0.75) | 300000 |
| Hard Goal | 20000 (0.067\|0.60) | 5000 (0.017\|0.55) | 300000 |
| Platform | 10000 (0.05\|0.98) | 5000 (0.025\|0.96) | 200000 |
| Catch Point | 100000 (0.1\|0.90) | 20000 (0.02\|0.82) | 1000000 |
| Hard Move (n=4) | 100000 (0.1\|0.93) | 20000 (0.02\|0.91) | 1000000 |
| Hard Move (n=6) | 100000 (0.1\|0.92) | 20000 (0.02\|0.92) | 1000000 |
| Hard Move (n=8) | 100000 (0.1\|0.89) | 20000 (0.02\|0.83) | 1000000 |
| Hard Move (n=10) | 100000 (0.1\|0.75) | 20000 (0.02\|0.70) | 1000000 |

Table 7: The exact number of samples used in warm-up stage training in different environments. The column of 'original' denotes what is done in our experiments; the column of 'new' denotes additional experiments we conduct with fewer warm-up samples (and proportionally fewer warm-up training). For each entry $x(\mathbf{y}|\mathbf{z})$, $x$ is the number of samples (environment steps), $\mathbf{y}$ denotes the percentage $\frac{\text{number of warm-up environment steps}}{\text{number of total environment steps during the training process}}$, and $\mathbf{z}$ denotes the corresponding performance of HyAR-TD3 as evaluated in Tab. 2. **Conclusion:** The numbers of warm-up environment steps are about $\mathbf{5\%-10\%}$ of the total environment steps in our original experiments. The number of warm-up environment steps can be further reduced by at most 80% off (thus leading to $< \mathbf{3\%}$ of the total environment steps) while comparable performance of our algorithm remains.

# E    ADDITIONAL DISCUSSION ON DISCRETE-CONTINUOUS HYBRID ACTION SPACE

To the best of our knowledge, most RL problems with discrete-continuous hybrid action spaces can be formulated by Parameterized Action MDP (PAMDP) described in Sec. 2. These hybrid action spaces can be roughly divided into two categories: structured hybrid action space (mainly considered in our paper) and non-structured hybrid action space. In non-structured hybrid action space, there is no dependence among different parts of hybrid action. For a mixed type of the above two, we may conclude it in the case of structured hybrid action space.

To be specific, learning with non-structured hybrid action spaces can be viewed as a special case of PAMDP. It can be similarly formulated with the definition of discrete action space $\mathcal{K}$ (p.s., this can be a joint discrete action space when there are multiple discrete dimensions) and the definition of

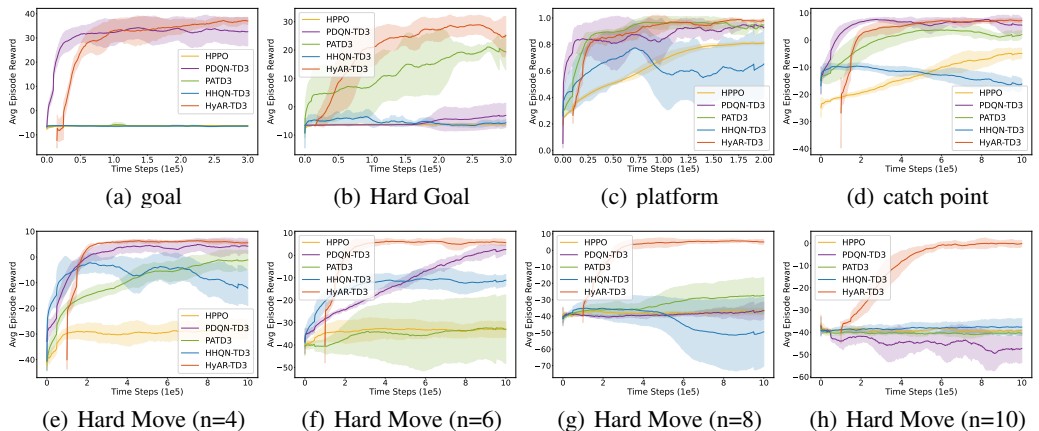

Figure 12: Another view of the learning curves shown in Fig. 5 where the number of samples used in warm-up training (i.e., stage 1 in Algorithm 1) is also counted for HyAR. Comparisons of algorithms in different environments. The x- and y-axis denote the environment steps and average episode reward over recent 100 episodes. The results are averaged using 5 runs, while the solid line and shaded represent the mean value and a standard deviation respectively.

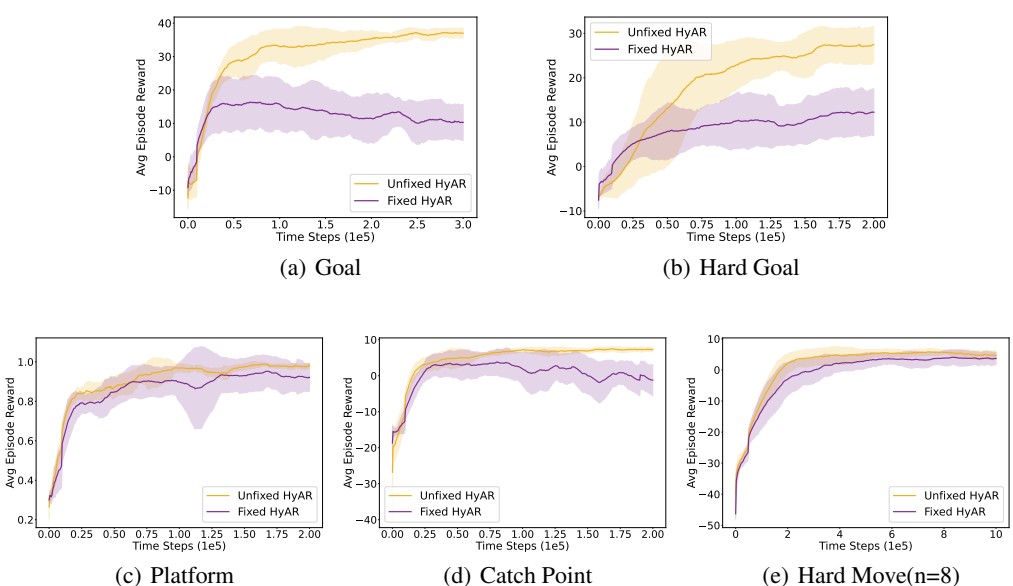

Figure 13: Comparison between HyAR-TD3 (denoted by **Unfixed HyAR**) and HyAR-TD3 with fixed hybrid action representation space trained in warm-up stage (denoted by **Fixed HyAR**) in different environments. The x- and y-axis denote the environment steps and average episode reward over recent 100 episodes. The results are averaged using 5 runs, while the solid line and shaded represent the mean value and a standard deviation respectively. **Conclusion:** Unfixed HyAR outperforms Fixed HyAR across all the environments; while Fixed HyAR performs very poorly in *Goal* and *Hard Goal*. We conjecture that in these environments, random policy is quite limited in collecting effective and meaning hybrid actions in these environments thus the learned fixed representation space is not able to support the emergence of effective latent policy.

a uniform continuous action space $\mathcal{X}$ rather than $\mathcal{X}_k$ since there is no structural dependence in this case. A few examples and analogies can be found in Robotic control (Neunert et al., 2019) where several gears or switchers need to be selected in addition to other continuous control parameters, or

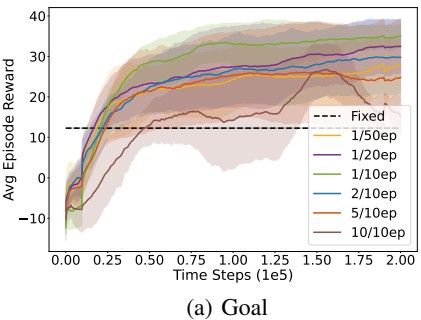
(a) Goal

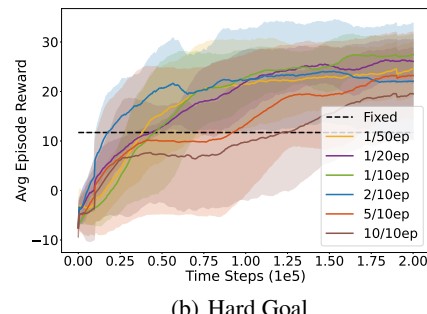
(b) Hard Goal

Figure 14: The effects of different choices of training frequency of subsequent representation training of HyAR-TD3 in the environments *Goal, Hard Goal*. For example, $1/10$ep denotes that the hybrid action representation is trained every **10 ep**isodes for **1** batches with batch size 64. The black dashed lines denote the convergence results of Fixed HyAR in Figure 13. The x- and y-axis denote the environment steps and average episode reward over recent 100 episodes. The results are averaged using 5 runs, while the solid line and shaded represent the mean value and a standard deviation respectively. **Conclusion:** all configurations of subsequent training outperform Fixed HyAR; while a moderate training frequency works best.

several dimensions of continuous parameters are discretized. In this view, our algorithm proposed under PAMDP is applicable to both structured and non-structured hybrid action spaces.

However, discrete-continuous hybrid action spaces are not well studied in DRL. Most existing works evaluate proposed algorithms in relatively low-dimensional and simple environments. Such environments may reflect some significant nature of discrete-continuous hybrid action control problem yet are far away from the scale and complexity of real-world scenarios, e.g., more complex robots with natural discrete and (dependent) continuous control (beyond the basic robot locomotion in like popular MuJoCo RL suite), RL-based software testing (Zheng et al., 2021b; 2019), game AI generation (Shen et al., 2020), recommendation systems (e.g., e-shop, music, news), and industrial production (e.g., production process control & smart grid control) (Sun et al., 2020; Zheng et al., 2021c). We believe that the emergence of such environments will benefit the community of discrete-continuous hybrid action RL study. To evaluate and develop our algorithm on multiagent scenarios (Hao et al., 2022; Zheng et al., 2021a; 2018) with more complex and practical hybrid action spaces is one of our main future directions.

