# OpenReview forum: "HyAR: Addressing Discrete-Continuous Action Reinforcement Learning via Hybrid Action Representation"
_ICLR.cc/2022/Conference — ICLR 2022 Poster_

### Official Review · Reviewer_JLuH · 2021-10-26

**Correctness:** 4
**Technical Novelty And Significance:** 3
**Empirical Novelty And Significance:** 3
**Recommendation:** 8
**Confidence:** 1

**Main Review:**

The concept of latent actions is very interesting. In the latent space, both discrete and continuous actions in the original space are represented by continuous vectors. This is based on the prior work on action representation learning for RL, but this work uses it to solve the difficulty of handling hybrid actions.

In addition, the decoder predicts the difference between current and next states (s and s'). This may be a small addition, but it looks reasonable to make the representations discriminative. This can be done in an unsupervised manner, so it doesn't introduce any additional cost of labeling.

This is how to deal with the hybrid actions, and it can be used in any RL methods by replacing actions with the proposed latent actions. This means that the method has a variety of potential applications.

A concern is that among the four factors in Table 1, stationarity is not explicitly discussed in the paper.


**Summary Of The Paper:**

This paper proposes to handle both continuous and discrete actions in reinforcement learning. Typical MDP suggests actions in either (not both) continuous or discrete action spaces, however in more general cases hybrid action spaces are needed. The proposed method solves this problem by using the latent space and the latent action. Instead of using raw actions, which are continuous and discrete, actions are embedded into a latent space, in which all are continuous. By decoding the latent action to actions in the original hybrid space, the whole process can be seamlessly integrated into any RL algorithms.
Experiments show that the proposed method method combined with TD3 and DDPG performs significantly better.


**Summary Of The Review:**

Overall the paper is well organized and written, and the provided code ensures the reproducibility.

---

> ### Author Response · Authors · 2021-11-16
> **Initial Response to Reviewer JLuH**
>
> 1. [Re: Discussion on Stationarity]
>
> We appreciate the reviewer's comments and we have added more discussion on the stationarity of PADDPG, PDQN, HPPO in the introduction for better clarity.
>
> The stationarity property denotes the stationarity in policy learning of discrete part and continuous part for an hybrid action algorithm.
> We discuss the non-stationarity of HHQN in the introduction (below Figure 1) and the stationarity of our algorithm in Section 3.1 (above Equation 3).
>
> In this sense, we consider PADDPG, PDQN, and HPPO to be stationary (as shown in Table 1) because they all learn a single overall policy and value function for hybrid action space thanks to their particular structures.
> This can also be considered as a joint policy learning in the view of a two-agent cooperative game as discussed for HHQN in the introduction, thus freeing the non-stationarity of individual learning.

---

### Official Review · Reviewer_Am1g · 2021-10-29

**Correctness:** 4
**Technical Novelty And Significance:** 3
**Empirical Novelty And Significance:** 3
**Recommendation:** 6
**Confidence:** 4

**Main Review:**

**Strengths**

- The paper is well written and easy to follow. The problem at hand is very interesting and the proposed solution meaningful. Some works, like AlphaStar had already tackled hybrid action spaces (although little detail is given in the corresponding paper), but their solution was very ad-hoc to the StarCraft problem, while the authors of this paper propose a very generic solution that could be applied to any (? I am not 100% sure about this, cf my question later) hybrid-action space problem.
- The experimental evaluation of the method is very complete, the ablations are convincing and enlightening the importance of the different components of the method.
- The authors made the effort to implement “enhanced” versions of the baselines (td3 based rather and ddpg based), which makes the comparisons fairer.


**Weaknesses**
- Method:
    - Is the method applicable to any hybrid action spaces? Is there not a underlying assumption that the continuous part of the action depends on the discrete part but not the other way around? Could you think about such examples?
    - For the sake of mathematical correctness, I think the function predicting $\hat{\delta}$ should not be named the same as the decoder as they are different, although some parameters are shared. If you want to mathematically underline the fact that some parameters are shared, you could write $g \circ f$ and $h \circ f$.
    - The paragraph starting with “Our cascaded structure” is very hand-wavy and unclear to me. As far I understand, $L_{Dyn}$ is just an auxiliary loss helping the learning of a better representation. Am I correct? If yes, I encourage the authors to simplify this paragraph. If not, I would encourage the authors to explain why and enhance the explanations given in this paragraph.
    - “Reward agnostic data of environmental dynamics which is easy to obtain”. I argue this statement is too strong. In particular, for a lot of environments, random behaviors lead to an extremely small coverage of the state space.
    - I would like the authors to discuss the potential drawbacks of the two main tricks: LSC and RSC. For example, 1) are there environments/setups for which they would be limiting. 2) RSC requires more memory because you have to store both the original action and the embedded one in the replay buffer, and it needs more compute to calculate several times the embedding of a given action. I think it would be valuable for the work if the authors are more open on the limitations of the method.

- Related Work
    - I think one related work that would be worth discussing and maybe include in the baselines is Neunert et.al. (https://arxiv.org/abs/2001.00449).
    - The absence of the “Related Work” section does help positioning the paper with respect to the literature.

- Experiments
    - The experiments, although convincing, are all made in pretty low dimensional state and action spaces. It would be of course an additional strength to validate the method in a more complicated environment.
    - Although interesting ablations are done, I would have liked two more. 1) What happens if the embeddings are fixed after pretraining? 2)     - What happens if you can’t pretrain (because no data is available at this point)?
    - I encourage the authors not to talk in terms of “learning steps” or “episodes” because these are hyperparameter-dependent (e.g. batch size dependent) and environment dependent. I argue they should use in x-axis the number of environment steps as is done for most online methods.
    - HyAR has access to more data than the baselines (the 5000 episodes used in the pretraining are not used by baselines…). Could you try to use them for the baselines too? For example filling the replay buffer with them? Otherwise I don’t find the comparison fair.
    - I would like to see a paragraph (in Appendix) discussing how hyperparameters were tuned. In particular, I think it is important to understand how much effort was put into tuning the method as well as the baselines.
The values of the most important hyperparameters should be explicit in the main text like the choice of $d_1$ and $d_2$.

- Writing
    - The writing is mostly clear (except this paragraph I mentioned earlier). Yet, there are a few typos, e.g. ”Variantional” in the abstract. “embeds the the dependence” in the introduction. “$\gamma^l$” in 2.1.


**Summary Of The Paper:**

- The authors are interested in the problem of Reinforcement Learning with hybrid action spaces. More precisely, MDPs with “structured” action spaces for which an action corresponds to the choice of a discrete value and a continuous vector. One important point is that the “meaning” of the continuous vector is dependent on the choice of the discrete action.
In order to tackle this issue, the authors propose to 1) embed the discrete action using an embedding table into a vector space of dimension $d_1$. 2) Use a conditional VAE to embed the continuous action in a vector space of dimension $d_2$. 3) Train a continuous-action RL algorithm in the $d_1+d_2$ latent space. 4) Decode the action back to the original action-space to pass it to the environment.

- The authors also propose a number of tricks to enhance the performance of this embedding: 1) an auxiliary loss predicting environment dynamics. 2) a rescaling of the policy to avoid actions that would not be correctly decoded. 3) A relabeling technique to ensure that off-policy-ness added to the learning of the action embeddings does not hurt the learning.


**Summary Of The Review:**

This is a very interesting paper that could be excellent with a bit more transparency in the method’s limitations as well as a bit more care in the experimental validation. I would increase my score if authors can provide details on the limitations of the method and tackle the major points I mention in the "experiment" part.

================== POST REBUTTAL 1 ================

I would like to thank the reviewers for the detailed answers and the revision of the manuscript.

The additional data provided by the authors in Table 7 and Figure 12 provide convincing proofs of the method's efficiency that were lacking (or at least that we could doubt) before the revision.
The additional ablation in Figure 13 brings nice insights on the methods. I even personnally would prefer to see it in the main text in place of the t-SNE visualizations as I find it more informative (but it is also fine like this!).

Insisting on these minor points:
I would rather not have footnote 2 and be more precise in the text.
I would also detail in the HP section of the appendix HOW hyperparemeters were selected and not only which ones were selected (even if the sentence just says that not much tuning was done, I think it is important.)

I am now recommending acceptance and raising my score to a 6 (it could be a 7 if it existed).
The work would be a clear 8 with an additional experiment on a more challenging and less toy-ish environment.
Yet, I understand it can be left for future work.

---

> ### Author Response · Authors · 2021-11-15
> **Initial Response to Reviewer Am1g - Part 1: Questions on the Experiments**
>
> 1. [Re: the use of "learning step" and "environment step"]
>
> After a careful check, we found we $\textbf{misused}$ the term "learning steps" in our paper (mainly in Figure 5 and Figure 7 and some other places where we use this term).
> $\textbf{In fact, the x-axes in Figure 5 and Figure 7 denote the environment steps,}$ i.e., number of agent-environment interaction samples used during the online training process.
> In all experiments, our comparison are conducted based on environment steps rather than hyperparameter-dependent terms.
>
> We appreciate the reviewer for pointing this term mistake and we have corrected all associated places in our revision.
>
> &nbsp;
>
> 2. [Re: Questions on the warm-up hybrid action representation training: 1) "What happens if you can’t pretrain (because no data is available at this point)?" 2) The exact number of samples used in warm-up training and the sample-fair comparison 3) "What happens if the embeddings are fixed after pretraining?"]
>
> We appreciate the reviewer for raising the questions and
> we provided additional experiments to make a better convey of the effect and implementation of warm-up representation training.
>
> First, for Question 1),
> we perform the warm-up representation training (i.e., stage 1 training as lines 4-6 in Algorithm 1) at the beginning of online training process in our implementation.
> This is similar to the short stage of random sample collection before policy and value function training as conventionally done for many online RL algorithms (e.g., TD3).
> Moreover, our warm-up representation training is light-weighted,
> since the number of samples used in this stage is only $\textbf{5\\% - 10\\%}$ of the total online environment steps (i.e., the x-axis of Figure 5) in our experiments.
> Therefore, our warm-up representation training is different from a thorough pre-training often seen in representation learning works.
>
> Next, for Question 2),
> we provide some additional results for this in our revision:
> - We provide the exact number of samples used in stage 1 in each environment in Table 7, about $\textbf{5\\% - 10\\%}$ of the total environment steps.
> - We also provide Figure 12 in the appendix for another view of the learning curves where the number of samples used in stage 1 is also counted for HyAR, i.e., a sample-fair comparison.
> - Moreover, we also conducted some experiments to further reduce the number of samples used in stage 1 (at most 80\% off). See the results in Table 7. We found that HyAR can achieve comparable performance with $< \textbf{3\\%}$ samples of the total environment steps.
>
> Then, for Question 3), we conducted additional experiments to demonstrate the necessity of the subsequent representation training after the warm-up stage.
> In specific, we compare HyAR-TD3 and HyAR-TD3 with fixed hybrid action representation space trained in stage 1 (called $\textbf{Fixed HyAR}$ below) in all our environments (see results in Figure 13 in our revision).
> We can observe that HyAR outperforms Fixed HyAR across all the environments, and meanwhile, Fixed HyAR performs poorly in $\textit{Goal}$ and $\textit{Hard Goal}$.
> This finding is reasonable since a random policy is quite limited in collecting effective and meaningful hybrid actions; thus, the learned fixed representation space cannot support the emergence of effective latent policy.
>
> &nbsp;
>
> 3. [Re: Hyperparameters tuning, especially the choice of $d_1$ and $d_2$]
>
> Overall, for HyAR, we did not perform exhausting hyperparameter tuning in our experiments. Thus, there remains much room for improving the learning performance of HyAR further when more rigorous tuning is adopted.
>
> For the major hyperparameters of HyAR, e.g., $d_1,d_2$ (we adopt the same dimensionality for both of them), LSC select range, auxiliary loss balancing weight $\beta$,
> we provide detail learning curves of different choices in Figures 8 \& 9 and Table 6.
> For the hyperparameters of latent policy learning (i.e., DDPG, TD3), we follow the corresponding conventions usually adopted in continuous control.
>
> To be more specific about $d_1,d_2$, we tried $d_1 = d_2$ from a candidate increasing dimensionality sequence $\{3, 6, 12, ...\}$ and found $d_1 = d_2 = 6$ works well in all our experiments.
> The insight for designing the candidate sequence is that: the proper dimensionality of latent action representation may be comparable (or more compact) to the dimensionality of state (ranging from 4 to 17 dimensions in different environments in our experiments). This is because the latent action representation should reflect the semantics of original hybrid action, i.e., the state residual.

---

> ### Author Response · Authors · 2021-11-15
> **Initial Response to Reviewer Am1g - Part 2: Questions and Discussions on Mothodology and Others (1)**
>
> 1. [Re: Question on more possible hybrid action spaces: "Is the method applicable to any hybrid action spaces? Is there not a underlying assumption that the continuous part of the action depends on the discrete part but not the other way around? Could you think about such examples?"]
>
> For the opposite case (the discrete part of hybrid action depends on the continuous part) that the reviewer questioned, we think this can exist theoretically, yet we can not come up with a representative scenario at present.
> For now, to the best of our knowledge, most RL problems with discrete-continuous hybrid action spaces can be formulated by Parameterized Action MDP (PAMDP described in Sec.~2.2),
> including both structured hybrid action space (mainly considered in our paper) and non-structured hybrid action space (as assumed in the related work Neunert et.al., 2020 mentioned by the reviewer).
> Therefore, our algorithm proposed under PAMDP is applicable to both structured and non-structured hybrid action spaces.
>
> To be specific, learning with non-structured hybrid action spaces can be viewed as a special case of PAMDP where no
> dependence exists among different parts of hybrid action.
> Similarly, this can be formulated with the definition of discrete action space $\mathcal{K}$
> (p.s., this can be a joint discrete action space when there are multiple discrete dimensions)
> and the definition of
> a uniform continuous action space $\mathcal{X}$ rather than $\mathcal{X}_{k}$ since there is no structural dependence in this case.
> A few examples and analogies can be found in Robotic control where several gears or switchers need to be selected in addition to other continuous control parameters,
> or several dimensions of continuous parameters are discretized (as done in the related work mentioned by the reviewer).
>
> &nbsp;
>
> 2. [Re: Clarification to the reviewer's misunderstanding of the cascaded structure]
>
> The reviewer's understanding about the auxiliary loss $L_{\text{Dyn}}$ is correct, however, the reviewer's understanding about the "cascaded structure" may be incorrect.
>
> We would like to clarify that the "cascaded structure" denotes the choice of $\textbf{network structure}$ for the implementation of the auxiliary loss $L_{\text{Dyn}}$, rather than the auxiliary loss $L_{\text{Dyn}}$ itself.
> There are two choices of network structure we considered in our paper:
> the "cascaded structure" and the "parallel structure" (as the contrast in our ablation study provided in Table 6).
>
> In other words, the state residual prediction network illustrated in Figure 2 is "cascaded" rather than "in parallel" to the reconstruction network.
> As we note in our paper, the adoption of the cascaded structure is inspired by (Azabou et al., 2021).
> Azabou et al. demonstrate the advantage of the cascaded structure over the parallel one
> when multiple auxiliary tasks are leveraged to learn a shared representation,
> as also done in our work.
>
> We appreciate the reviewer's help in revealing such potential ambiguity here. We have polished our text to make this more accurate in our revision.
>
> &nbsp;
>
> 3. [Re: About reward agnostic data]
>
> The training of our hybrid action representation involves no rewards.
> Here we intend to say that reward-agnostic data could be convenient to obtain than reward-dependent data, since the reward signals are often non-trivial to design or collect.
> Such opinions are also recognized in some related studies on reward-agnostic self-supervised state representation learning [1,2,3].
>
> In our paper, we use random samples (can be $< 3\\%$ of total environment steps) for the warm-up representation training rather than a thorough pre-training, while later the representation is trained along with RL training based on the state-action samples encountered during the online learning process.
>
> We appreciate the reviewer for pointing out this and have rephrased the associated text to a more accurate expression in our revision.
>
> &nbsp;
>
> ------------
>
> Reference:
>
> [1] Adam Stooke, Kimin Lee, Pieter Abbeel, Michael Laskin. Decoupling Representation Learning from Reinforcement Learning. ICML 2021
>
> [2] Akram Erraqabi, Mingde Zhao, Marlos C. Machado, Yoshua Bengio, Sainbayar Sukhbaatar, Ludovic Denoyer, Alessandro Lazaric. Exploration-Driven Representation Learning in Reinforcement Learning.  Unsupervised Reinforcement Learning workshop, ICML 2021
>
> [3] Denis Yarats, Rob Fergus, Alessandro Lazaric, Lerrel Pinto. Reinforcement Learning with Prototypical Representations. ICML 2021

---

> ### Author Response · Authors · 2021-11-15
> **Initial Response to Reviewer Am1g - Part 3: Questions and Discussions on Mothodology and Others (2)**
>
> 1. [Re: More discussion on the potential limitations and improvements of LSC and RSC]
>
> We believe the ideas of LSC and RSC are general.
> In our work, both LSC and RSC are easy to implement and empirically demonstrated to be effective.
> Also, we believe there is room for these two mechanisms to improve further in more challenging cases.
>
>
> For LSC, one potential drawback may be that the central range constraint on per dimension adopted in our work,
> is suboptimal to the case if the distribution of learned action representation of original hybrid actions is multi-modal.
> In such cases, the central range constraint may be less effective and can be further improved.
>
> For RSC, it does require additional memory and computation cost.
> However, in our experiments, we found that such memory and computation cost is relatively negligible.
> The dimensionality of each original hybrid action sample is relatively small compared to the state sample stored;
> the additional wall-clock time cost of HyAR (with RSC) to HyAR without RSC is about $\textbf{6\\%}$ (tested in $\textit{Goal}$).
> We also agree that in some storage- and computation-sensitive problems, this may need more considerations.
>
> As suggested, we have enriched the discussion in the revision and we will consider the above potential improvements as future work.
>
> &nbsp;
>
> 2. [Re: About the related work mentioned, Neunert et.al. (https://arxiv.org/abs/2001.00449)]
>
> We appreciate the reviewer for recommending the related work.
> We will add the discussion in the revision and compare it in our experiments if we can reproduce its results (we have not found the open-source code of this paper).
>
> After reading this work, we believe there exists a critical difference between their and our approaches. Their work assumes that all dimensions of both discrete and continuous actions are $\textbf{independent of each other}$, while ours does not. Such an assumption may narrow the algorithm's generalization. For instance, most experimental environments in their work are customized by discretizing continuous actions, which do not well reflect the nature of hybrid action space problems like $\textit{Goal}$ problem.
>
> Besides, in our understanding, the related work and HPPO (the baseline in our experiments) adopt the same hybrid policy modeling/representation fashion
> (with the major difference at the policy optimization algorithm used, i.e., PPO v.s. MPO), thus very likely having similar performance.
>
> As suggested, we will try to reproduce it to enrich our experimental comparisons.
>
> &nbsp;
>
> 3. [Re: The difference between decoder and prediction network]
>
> We agree with the reviewer that it may be misleading to conclude the prediction network in the square denoted by text "decoder" as in Figure 2.
>
> We have added a footnote to clarify this in our revision and we will consider to repair this as suggested for better clarity.
>
> &nbsp;
>
> 4. [Re: Writing]
>
> Thanks for the reviewer's comments and we have amended them in our revision.
>
> &nbsp;

---

> > ### Comment · Reviewer_Am1g · 2021-11-16
> > **Answer**
> >
> > Thanks for the very detailed answer and the revision. Please check my detailed answer below my original review!

---

> ### Author Response · Authors · 2021-11-16
> **Response to POST REBUTTAL 1 Comments from Reviewer Am1g**
>
> We appreciate the reviewer's quick response very much!
>
> We highly agree with the reviewer's insisted suggestions. We add the footnote in our current revision only for a temporary clarification for the convenience of review discussion. Careful amendment on this point and other valuable suggestions will be seriously made and considered in our later revision.

---

### Official Review · Reviewer_TSgb · 2021-10-31

**Correctness:** 3
**Technical Novelty And Significance:** 3
**Empirical Novelty And Significance:** 4
**Recommendation:** 8
**Confidence:** 4

**Main Review:**

Pros:

- The paper is overall well-written and easy to follow. The being studied problem and mathematics are clearly defined.

- The proposed method is well motivated and well designed. In particular, two mechanisms are proposed (LSC, RSC) to alleviate the corresponding issues, which I feel important and should be described in more details in the main texts.

- The experimental results are convicinble and comprehensive. Abalation studies is well-conducted. Implemetation details are clearly explained.

Cons:

- It seems HyAR is relatively sensitive to the choice of hyper-parameters (Table 6). It remains unclear how to select good hyper-parameters rather than grid/random search.

- For the representation shift problem, although the proposed RSC mechanism showed empirical performance gain, I feel that the problem is not "resolved", but only "alleviated". Therefore, to say HyAR is fully stationary is a bit overclaimed in my opinion (Table 1).

Minor problems:

- In the definition of Q function in 2.1: \gamma^l  --> \gamma^t

- VAE reference: bayes -> {B}ayes, similar to t-SNE, etc.


**Summary Of The Paper:**

The authors proposed a novel framework for hybrid (discrete and continuous) action RL, called Hybrid Action Representation (HyAR). The main idea is to take advantage of representation learning in Deep RL to encode hybrid action in a conbtinuous latent vector space. Then any RL algorithm for continuous action space can be used to train the agent. Importantly, to mitigate the practical problems of unreliability and representation shift of the learned action encoding, the paper proposed two techniques LSC and RSC, which were empirically shown to be effective.

Experiments in various hybrid action tasks demonstrated that the proposed HyAR method contribute to significant performance gain to the baseline TD3 and DDPG models, as well as previous methods. The authors further performaned an ablation investigation to justify their claims. Moreover, the t-SNE visualization of learned actions representation shows an relatively interpretable latent encoding.

**Summary Of The Review:**

The paper proposed a method to solve hybrid action control problems by encoding action to a continuous latent space via representation learning. The motivation is sound and the writting is clear. The technical contribution is significant. The experiments are comprehensive and support the paper's major claims. Overall, I recommend acceptance.

================== post-rebuttal ==================

Although I believe this work has some limitations that can be improved in the future, I feel the contributions of the current paper are already significant and I vote for acceptance.

---

> ### Author Response · Authors · 2021-11-16
> **Initial Response to Reviewer TSgb**
>
> 1. [Re: Discussion on the choice of hyper-parameters of HyAR]
>
> For HyAR, we did not perform exhausting hyperparameter tuning in our experiments.
> We detail the process of our hyperparameter tuning below.
>
>
> Overall, for the major hyperparameters of HyAR, i.e., $d_1,d_2$, LSC select range $c$, auxiliary loss balancing weight $\beta$,
> we provide the learning curves of different choices of each of them in Figures 8 \& 9 and Table 6.
> For the hyperparameters of latent policy learning (i.e., DDPG, TD3), we follow the corresponding conventions usually adopted in continuous control.
>
> To be more specific:
>
> - For $d_1,d_2$, we tried $d_1 = d_2$ from a candidate increasing dimensionality sequence $\\{3, 6, 12, ...\\}$ and found $d_1 = d_2 = 6$ works well in all our experiments.
> Results of different choices are shown in Figure 9 (d) \& (h) and Table 6.
> The insight for designing the candidate sequence is that: the proper dimensionality of latent action representation may be comparable (or more compact) to the dimensionality of state (ranging from 4 to 17 dimensions in different environments in our experiments).
> This is because the latent action representation should reflect the semantics of the original hybrid action, i.e., the state residual.
>
> - For LSC select range $c$, we started from no constraint (i.e., $c=100$) and then gradually strengthened the constraint, resulting a decreasing range sequence $\\{100, 96, 90, 80, ...\\}$.
> Results of different choices are shown in Figure 9 (c) \& (g) and Table 6.
> We chose $c=96$ as default because it works well in all our environments, while no constraint or stronger constraint hinders learning performance less or more in different environments.
>
> - For auxiliary loss balancing weight $\beta$, we provide the learning curves of different choices in Figure 8(c) and some discussion in Appendix C.2.
> We searched the candidate set $\\{0.1, 1, 5, 10, 20\\}$ with increasing scales, and we found that $\beta=10$ works well in all our environments.
> We also found that the learning performance gradually improves with the increase of weight $\beta$, reaches the best when $\beta=10$ since the dynamics predictive representation goes into effect well;
> then the performance goes down when $\beta=20$ because the over-weighting of dynamics prediction loss may dominate the VAE loss and hinder the reconstruction quality.
>
>
> We appreciate the reviewer's comments, and we have added more discussion about the hyperparameter choices in our revision.
> We agree that further investigation will facilitate efficient hyperparameter tuning, and we believe there remains much room for improving the learning performance of HyAR further if more sophisticated hyperparameter tuning is adopted.
>
>
> &nbsp;
>
> 2. [Re: Clarification on Stationarity in Table 1]
>
>
> We appreciate the reviewer for pointing out this, which helps us enrich our discussion on stationarity.
>
> The stationarity property in Table 1 denotes the stationarity in policy learning of discrete action and continuous action.
> In other words, it denotes whether the policy learning of either discrete part or continuous part (in an joint or decoupled fashion depends on different algorithms) may cause non-stationarity to the learning process of the other part.
> Here, we did not intend to involve the stationarity of representation learning (since prior works do not learn hybrid action representation) or other more general perspective on stationarity (e.g., off-policy RL non-stationarity).
> In this sense, we discuss the non-stationarity of HHQN in the introduction (below Figure 1) and the stationarity of our algorithm in Section 3.1 (above Equation 3).
> In addition, we have added some more discussion on the stationarity of PADDPG, PDQN, and HPPO in the introduction for better clarity in our revision.
>
> We agree with the reviewer's opinion on RSC, and we have checked the associated text and made slight amendments in our revision as suggested.
>
> &nbsp;
>
> 3. [Re: Minor problems]
>
> Thanks for the reviewer's comments and we have amended them in our revision.

---

> > ### Comment · Reviewer_TSgb · 2021-11-21
> > **Response**
> >
> > I appreciate the authors' efforts to revsolve my concerns. I maintain my recommendation of acceptance.

---

### Official Review · Reviewer_9YW6 · 2021-11-03

**Correctness:** 4
**Technical Novelty And Significance:** 3
**Empirical Novelty And Significance:** 3
**Recommendation:** 8
**Confidence:** 4

**Main Review:**

Strengths:
- Novel approach to handle mixed discrete-continuous action spaces
- Complete evaluation with comparison to multiple baselines

Weaknesses:
- The paper focuses on a specific type of mixed discrete-continuous action space. While this setup is common in robotics and other ML problems, other setups with mixed spaces exist, for example, robots with continuous and discrete joints (e.g. a binary “close/open” gripper action). To avoid confusion I would make this very clear in the intro/abstract
- Understanding the contribution of LSC and RSC is critical. I would include more about it in the main text
- It would be interesting to apply it to more complex robotics problems (e.g. robot control) to see the limits of the solution


**Summary Of The Paper:**

The paper presents an approach to handle action spaces with discrete selections and continuous parameters within each selection. This is a common setup in robotics, where agents need to select a skill to apply, and then the parameters to instantiate that skill (e.g. MOVE skill and where to move). The paper proposes an embedding space where the discrete component is mapped through a nearest neighbor selection, and the continuous component is mapped with a decoder conditioned on the embedding of the discrete component. The work includes additional modifications to improve policy learning: first, a method to select actions only in the region confidently represented in the mapped action space, and second, a method to remap the actions every time the mapping changes during training to keep the experience replay buffer “valid”. The method to handle heterogeneous discrete-continuous action spaces and the two extensions for RL training are evaluated on several simple interactive tasks that were used before in related work papers proposing methods for mixed discrete-continuous action spaces. The results indicate an improvement over existing methods.

**Summary Of The Review:**

I would recommend accepting the paper for the conference. The text can be streamlined (there is a couple of typos) but the results are pretty solid, the problem is relevant and the idea is novel.

---

> ### Author Response · Authors · 2021-11-16
> **Initial Response to Reviewer 9YW6**
>
> 1. [Re: Discussion on more possible hybrid action spaces]
>
>
> To the best of our knowledge, most RL problems with discrete-continuous hybrid action spaces can be formulated by Parameterized Action MDP (PAMDP) described in Sec.~2.2,
> including both structured hybrid action space (mainly considered in our paper) and non-structured hybrid action space (in which no dependence among different parts of hybrid action exists; a few examples provided in the related work Neunert et.al. https://arxiv.org/abs/2001.00449 mentioned by Reviewer Am1g).
> For a mixed type of the above two, we may conclude it in the case of structured hybrid action space.
> Therefore, our algorithm proposed under PAMDP is applicable to both structured and non-structured hybrid action spaces.
>
>
> To be specific, learning with non-structured hybrid action spaces can be viewed as a special case of PAMDP,
> which can be similarly formulated with the definition of discrete action space $\mathcal{K}$
> (p.s., this can be a joint discrete action space when there are multiple discrete dimensions like for multiple grippers)
> and the definition of
> a uniform continuous action space $\mathcal{X}$ rather than $\mathcal{X}_{k}$ since there is no structural dependence in this case.
> A few examples and analogies can be found in Robotic control where several gears or switchers need to be selected in addition to other continuous control parameters,
> or several dimensions of continuous parameters are discretized (as done in the related work mentioned above).
>
>
> We appreciate the reviewer's comments and we have added some discussion on this in Appendix E in our revision to enrich our paper.
>
> &nbsp;
>
> 2. [Re: Future work on more complex hybrid action space experimental environments]
>
> We agree with the reviewer.
> To evaluate and develop our algorithm on more complex hybrid action spaces is one of our main future directions.
>
> As suggested by the reviewer, more complex robotics problems will be a good choice, yet we do not currently find some open-source and RL-standard robotics environments.
> One potential alternative is to customize the popular robotics environments conventionally used for continuous control, e.g., MuJoCo.
> The simplest way is to discretize several continuous dimensions.
> However, such modifications neglect the nature of structural dependence, which is significant in real-world hybrid action space problems.
>
> We consider that the appropriate and more meaningful development of hybrid action RL environments needs further thought.
> This may be done by taking inspiration from practical scenarios,
> for example, more complex robots with natural discrete and (dependent) continuous control, similar to the case mentioned by the reviewer, rather than the basic robot locomotion in the popular MuJoCo RL suite.
> Other possible scenarios can be recommendation systems (e.g., e-shop, music, news) and industrial production (e.g., production process control).
>
> We are looking forward to the emergence of such environments, which will benefit the community of discrete-continuous hybrid action RL study.
>
> &nbsp;
>
> 3. [Re: Suggestion on including more about LSC and RSC in the main text]
>
> We appreciate the reviewer's constructive comments.
> We will plan to streamline the main text of our paper and include more details and discussion on LSC and RSC in our later version.

---

> > ### Comment · Reviewer_9YW6 · 2021-11-22
> > **Post-rebuttal**
> >
> > Thank you for your responses and the improvements to my comments and the comments from my fellow reviewers. I've raised my score, although like reviewer Am1g, I'd prefer to set a 7 because of the lack of more complex experimental evaluation to test the limits of the proposed approach.

---

### Author Response · Authors · 2021-11-15
**We appreciate all the reviewers' careful and valuable comments**

We appreciate all the reviewers' careful and valuable comments.
Individual responses and our revision will be provided soon.

We are looking forward to the following inspiring discussions.

---

### Author Response · Authors · 2021-11-15
**Updates Made in Our Revisions**

We have uploaded a revision in which all modifications are colored in orange.

Major updates made in the $\textbf{first}$ revision are summarized below:

- The misuse of term "learning step" is checked and amended to be "environment step" at associated places for correctness.

- Additional experimental results are provided, including:

  - The exact numbers of samples used in the warm-up representation training on different environments in Table 7, i.e., 5\% - 10\% of the total online environment steps.

  - We also provide in Figure 12 for a sample-fair comparison of the learning curves where the number of samples used in the warm-up stage is also counted for HyAR.

  - We conducted additional experiments to further reduce the number of samples used in the warm-up stage with at most an 80\% off.
See the colored results in Table 7. It shows that HyAR can achieve comparable performance with $<$ 3\% samples of the total environment steps.

  - We compare HyAR-TD3 and HyAR-TD3 with fixed hybrid action representation space trained in the warm-up stage in all our environments, to demonstrate the necessity of the subsequent representation training after the warm-up stage.

- More discussions on our algorithm and some rephrasing are made.

- Some other experimental details are added.

--------

Updates made in the $\textbf{second}$ revision uploaded:

1) Several places in the appendix are slightly modified for accuracy and clarity (e.g., the redundant parameter $n$ of LSC originally used in Appendix A and Table 6 is removed to avoid ambiguity).

2) Layouts of the figures and tables in the appendix are slightly adjusted.

3) Some additional discussions are added (e.g., hyperparameter choices, more discussion on discrete-continuous hybrid action spaces).

--------

Updates made in the $\textbf{third}$ revision uploaded:

1) We adopted the suggestions of Reviewer Am1g and made small modifications to express the reconstruction network (i.e., decoder) and dynamics prediction network with $g \circ f$ and $h \circ f$ for better mathematical clarity. In addition, we update Figure 2 by adding two dashed lines and text for a more accurate illustration of the reconstruction network and dynamics prediction network.

2) We provide additional experimental results in Figure 14 in the appendix to demonstrate the effects of different choices of training frequency of the subsequent representation training. The results show that all configurations of subsequent training outperform using fixed representation trained in the warm-up stage; while a moderate training frequency works best. These results further completes the comparison we provide in Figure 13.

3) We polished some writings and amended several minor problems in the appendix.
--------

Further updates made in following revisions will be informed.

---

### Decision · Program_Chairs · 2022-01-20

**Decision:**

Accept (Poster)

**Comment:**

This paper proposes a new approach to solve mixed discrete-continuous action RL problems, based on embedding actions into a latent space so that standard continuous control algorithms (like TD3) can be applied. Experiments over standard discrete-continuous benchmarks demonstrate the superiority of the proposed approach vs. existing baselines.

There is overall a strong consensus of all reviewers towards acceptance, especially after the discussion period where the authors were able to submit several revisions addressing most of the questions and concerns raised in the original reviews, in particular w.r.t. the quality and relevance of the results.

I believe this submission could be improved along two axes though:

1. As pointed out by some reviewers, the current environments are somewhat simple. A more realistic robotic task for instance could be a good fit for such an algorithm. That being said, as the authors pointed out, this may require custom development due to the lack of existing public environment with the proper setup.

2. As a reviewer mentioned, there is no "Related Work" section, and although previous work is discussed in the Introduction, I consider that it remains limited, and more previous work should have been discussed. Here are some pointers regarding relevant work I am aware of:
- Hierarchical Approaches for Reinforcement Learning in Parameterized Action Space (https://arxiv.org/abs/1810.09656)
- Neural Ordinary Differential Equation Value Networks for Parametrized Action Spaces (https://openreview.net/forum?id=8WKd467B8H)
- Improving Action Branching for Deep Reinforcement Learning with A Multi-dimensional Hybrid Action Space (https://ipsj.ixsq.nii.ac.jp/ej/index.php?action=pages_view_main&active_action=repository_action_common_download&item_id=199976&item_no=1&attribute_id=1&file_no=1&page_id=13&block_id=8)
- Distributed Reinforcement Learning with Self-Play in Parameterized Action Space (https://cgdsss.github.io/pdf/SMC21_0324_MS.pdf)
- Discrete and Continuous Action Representation for Practical RL in Video Games (https://arxiv.org/abs/1912.11077)
- Multi-Pass Q-Networks for Deep Reinforcement Learning with Parameterised Action Spaces (https://arxiv.org/abs/1905.04388)

In particular, I believe the last one (MP-DQN) should have been one of the baselines, since it is supposed to be an improvement over the P-DQN algorithm (that is one of the baselines used here). I encourage the authors to try and incorporate it for the final version (at the very least, it should be cited).

In spite of these concerns, I still recommend acceptance since the combination of action space embedding with mixed discrete-continuous actions is novel and non-trivial, and the empirical validation is convincing enough in its current state.